# Stomata-targeted nanocarriers enhance plant defense against pathogen colonization

Suppanat Puangpathumanond[1], Heng Li Chee[2], Cansu Sevencan [1], Xin Yang [3], On Sun Lau [3,4] & Tedrick Thomas Salim Lew [1,4,5] ✉

Plant pathogens significantly threaten food security and agricultural sustainability, with climate change expected to exacerbate outbreaks. Despite these growing threats, current agrochemical delivery remains untargeted and inefficient. In this study, we develop **s**urface ligand-**e**ngineered **n**anoparticles for targeted **d**elivery to **s**tomata (SENDS), a nanocarrier system designed to target stomatal guard cells, which serve as key pathogen entry points into the plant apoplast. Our approach employs rational ligand engineering of porous nanoparticles, optimizing ligand orientation for efficient stomata targeting across different plant species. Foliar application of SENDS encapsulating an antimicrobial plant alkaloid reduces colonization of *Xanthomonas campestris*, a major crop pathogen, by 20-fold compared to untargeted nanocarriers. Quantitative assessment of stomatal aperture movement and photosynthetic performance confirms that SENDS enhance plant defense against invading pathogens without disrupting natural stomatal function. This nanobiotechnology approach provides a targeted strategy to improve plant disease resistance, offering new insights into nanocarrier design for more resilient and sustainable agriculture.

Plant pathogens and their associated diseases pose significant threats to agricultural productivity and global food security[1]. Crop yield losses attributed to these diseases amount to an estimated US$220 billion annually[2], highlighting their severe economic impact worldwide. As climate change intensifies, the risks associated with these pathogens are expected to rise[3,4]. This phenomenon is driven by several key factors related to climate change: enhanced pathogen proliferation and virulence, the emergence of new pathogen strains, and the expanding geographical range of existing pathogens[2,5–7]. These changes present considerable challenges for future agricultural production, with projections suggesting that any prospective gains in crop yields over the next 50 years may be nullified by the pressures of pathogen proliferation[8]. Recent advances in nanotechnology, specifically nanocarriers to deliver active agents with enhanced treatment efficacies, reduced toxicity, and limited leaching potential[9–14], have emerged as

promising solutions to address these challenges. Despite this progress, the precise delivery of these nanocarrier formulations to intended plant cells and organs remains challenging due to our limited understanding of plant-nanoparticle interactions, precluding rational nanoparticle design rules for targeted delivery. While studies in nanomedicine delivery have yielded promising strategies for targeted delivery to specific animal organs or tissues[15,16], similar design principles for precise nanocarrier delivery to plant organs or structures remain understudied and underdeveloped. Addressing these gaps would be essential to fully harness the potential of nanotechnology and strengthen agricultural resilience against plant pathogens in a rapidly changing climate.

Recent developments in elucidating fundamental plant-nanoparticle interactions largely focus on subcellular-level mechanisms. For instance, the lipid exchange envelope and penetration (LEEP)

[1]Department of Chemical and Biomolecular Engineering, National University of Singapore, Singapore, Singapore. [2]Institute of Materials Research and Engineering, Agency of Science, Technology and Research, Singapore, Singapore. [3]Department of Biological Sciences, National University of Singapore, Singapore, Singapore. [4]Research Centre for Sustainable Urban Farming, National University of Singapore, Singapore, Singapore. [5]NUS Environmental Research Institute, Singapore, Singapore. ✉e-mail: tedrick@nus.edu.sg

mechanism, along with various studies building on this knowledge, helps to describe the process of nanoparticle translocation past various subcellular structures of plants, from chloroplast and plant cell membranes to the pollen grains of flowering plants[17–19]. Studies based on the LEEP model illustrate the effect of size and zeta potential of various nanoparticles on their subcellular trafficking ability, while the effects of other physical properties, such as shape and stiffness, have also been found to affect nanoparticle internalization through alternative mechanisms such as endocytosis[20–22]. Besides tuning the physicochemical properties of nanoparticles, biorecognition strategies employing organelle-targeting peptides have been explored to target plant subcellular organelles, such as mitochondria and chloroplasts[23]. Targeted nanoparticle delivery to these subcellular organelles has enabled precise plant engineering applications previously unattainable, including manipulation of chloroplast redox status and mitochondrial genome engineering[23]. Beyond targeting subcellular organelles, strategies for directing nanoparticles to specialized plant cellular structures presents new opportunities in plant engineering. Targeted delivery using antibodies that recognize specific carbohydrates in the plant cell wall has been instrumental in studying cell wall structure and functionality[24,25]. This approach has been widely validated, as demonstrated by immunogold labeling in electron microscopy to visualize plant cell wall structure and composition[26,27]. Leveraging similar biorecognition strategies could enable precise nanoparticle delivery to important epidermal leaf structures, such as stomata, that are accessible through foliar application. This delivery method complements organelle-targeting strategies typically achieved through manual mechanical infiltration, thereby facilitating scalable precision delivery systems in agriculture.

Stomata are natural openings on the leaf epidermis formed by a pair of guard cells that are essential for regulating gas exchange during photosynthesis and managing plant stress responses. Their movements are intricately controlled by coordinated mechanisms involving plant hormones such as abscisic acid (ABA), reactive oxygen species (ROS) and signaling molecules[28–30]. The regulation of stomata movements allows plants to maximize gas exchange for photosynthesis while defending against stressors. Despite their critical functions in plant physiology, stomata also present a structural vulnerability, as they provide numerous openings on the epidermis serving as major entry points for pathogens into the plant tissue, thereby facilitating infection and promoting bacterial virulence[29]. Although stomata are equipped with mechanisms to sense pathogenic microbes and close the pore to limit pathogen entry, such self-defense mechanisms are often insufficient to fully prevent pathogen invasion and may be overwhelmed by aggressive or abundant pathogens[29,31]. Given these challenges, delivering nanoparticles specifically to guard cells of stomata is an attractive strategy to enhance plant defense by blocking pathogen entry at these biological gateways, without disrupting the natural physiology of stomatal movement. Despite earlier observations that functionalized gold nanoparticles and borate-zinc nanoformulations showed preferential localization in the stomata and trichomes[32,33], there is still a lack of studies on how stomata-targeted delivery of nanoparticles could affect plant fitness.

In this work, we develop a stomata-targeting nanocarrier platform, termed **s**urface ligand-**e**ngineered **n**anoparticles for targeted **d**elivery to **s**tomata (SENDS) and demonstrate its ability to enhance plant tolerance against pathogen infection (Fig. 1). The SENDS platform consists of a porous metal organic framework (MOF) nanoparticle core functionalized with modularly assembled protein ligands designed to target a specific polysaccharide on stomatal guard cells. Leveraging metal-histidine coordination chemistry and protein-protein binding, we achieve oriented bioconjugation of targeting ligands on the SENDS surface, improving the accessibility of the biorecognition element and maximizing stomata-targeting efficiency. Confocal and scanning electron microscopy confirm the stomata-

targeting capability of SENDS in five different plant species: *Arabidopsis thaliana*, *Brassica rapa* (subsp. *chinensis*) *Vicia faba*, *Oryza sativa* and *Hordeum vulgare*. After encapsulation of a natural antimicrobial plant alkaloid within the porous framework of SENDS, we show that the stomata-targeted delivery of SENDS augmented plant defense against pathogen colonization, with approximately 20-fold improvement in efficacy compared to untargeted nanoparticles. By specifically targeting stomata, key entry points for pathogens, SENDS disrupts pathogen colonization at its earliest stages, addressing a major vulnerability in plant defense mechanisms. Nanoparticle biocompatibility is further validated through quantifications of plant photosynthetic performance. Our findings highlight the critical role of plant-nanoparticle interactions in developing targeted nanocarrier systems, offering valuable insights for the design of nanomaterials to enhance agricultural sustainability and crop protection.

## Results
### Synthesis and characterization of SENDS
MOFs, a class of crystalline hybrid materials composed of metal ion clusters coordinated to organic ligands, was chosen as the nanoparticle core of SENDS due to their exceptionally high surface areas, tunable pore sizes, and chemical versatility, rendering them a suitable nanocarrier platform amenable to surface engineering and cargo encapsulation. Specifically, zeolitic imidazolate framework 8 (ZIF-8), composed of tetrahedrally-coordinated $Zn^{2+}$ ions and 2-methylimidazole, was selected for its proven ability to encapsulate and release various guest molecules[34–38], making it well suited for stomata-targeted agrochemical delivery. Additionally, $Zn^{2+}$ is biocompatible and generally benign to plants, with some studies suggesting secondary benefits as a micronutrient[34,39,40]. To endow ZIF-8 with stomata-targeting capability, targeting protein ligands were modularly assembled on the nanoparticle surface by leveraging its rich surface chemistry (Fig. 2A). The ZIF-8 core of SENDS was first synthesized under aqueous conditions at room temperature. Cetyltrimethylammonium bromide (CTAB) was used to modify the morphology of the ZIF-8 nanoparticles from their classical dodecahedral shape to a cubic shape (Fig. 2B)[41]. This modification was intended to increase the outer surface area-to-volume ratio of the particles, thereby allowing for the attachment of a higher number of stomata-targeting biomolecules (Supplementary Fig. 1F). Scanning electron micrographs (SEM) confirmed the cubic morphology of ZIF-8 with a Feret's diameter of approximately 225 nm (Supplementary Fig. 1A, B). Additionally, the powder X-ray diffraction (PXRD) diffractogram displayed characteristic peaks consistent with the simulated spectrum of ZIF-8, confirming its successful synthesis (Fig. 2C).

The stomata-targeting functionality is imparted to ZIF-8 through a modular assembly process which consists of two key steps (Fig. 2A): (i) bioconjugation of polyhistidine-tagged protein G (His-pG) to ZIF-8 and (ii) site-specific, oriented conjugation of arabinan-targeting LM6 immunoglobulin G (IgG) antibodies to His-pG. Arabinan is a pectic polysaccharide abundant in the cell wall of guard cells which plays a crucial role in regulating stomatal aperture by contributing to the flexibility and extensibility of the guard cell wall[42,43]. Targeting this molecule with arabinan-specific LM6 IgG therefore provides a biorecognition opportunity to target the stomata[44]. The first bioconjugation step was carried out by incubating a ZIF-8 dispersion with His-pG at room temperature. This conjugation is facilitated by Lewis acid-base interactions between the imidazole functional groups in the pG polyhistidine tags (Lewis base) and the coordinatively-unsaturated $Zn^{2+}$ sites (Lewis acid) on ZIF-8[45]. The high affinity of histidine towards transition metal ions, including $Zn^{2+}$, facilitates the resilience and efficiency of this binding interaction[46]. In the second step, IgG functionalization was performed by incubating His-pG functionalized ZIF-8 in a LM6 IgG solution at room temperature. During this process, IgG is optimally oriented on the nanoparticle surface by binding to His-pG

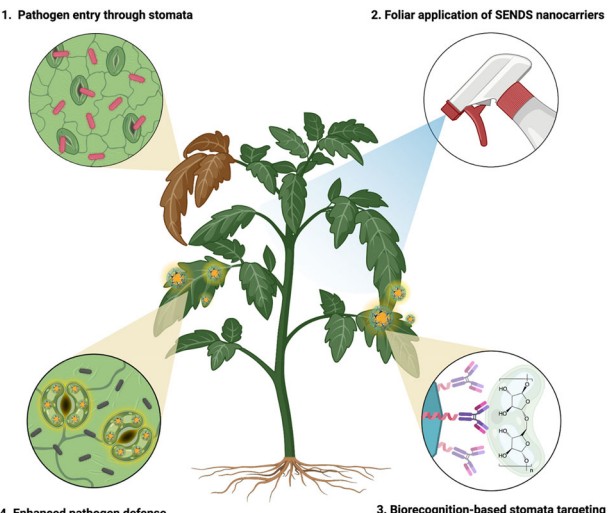

**Fig. 1 | Enhanced plant defense against pathogen colonization by foliar application of SENDS.** Stomata-targeting nanocarriers (SENDS) were specifically functionalized with a biorecognition moiety that facilitates high-efficiency targeting of stomatal guard cells. After foliar spraying, SENDS disrupted pathogen invasion through the stomata to improve plant defense against biotic stress. Created in BioRender. Lew, T.T.S (2025) https://BioRender.com/c54kfsb.

through high-affinity interactions between the Fc region of IgG and the C2 domain of His-pG ($K_D \sim 10\,nM$)[47], while leaving the Fab target recognition element of IgG exposed. Although studies have shown that IgG can be directly bound to ZIF-8[48], incorporating His-pG as an intermediate linker between IgG and ZIF-8 optimizes the orientation of the stomata-targeting IgG. This modular assembly approach increases the accessibility of the antigen-binding region of IgG on the surface of SENDS, maximizing its targeting capabilities.

To ascertain successful bioconjugation, SENDS were washed post-synthesis and incubated in Laemmli buffer to denature and strip the conjugated biomolecules from the surface. The biomolecules eluted off SENDS were then characterized using sodium dodecyl sulfate-polyacrylamide gel electrophoresis (SDS-PAGE) with Coomassie Blue staining. In contrast to plain ZIF-8 samples which showed no visible protein band, the lane corresponding to SENDS displayed strong protein bands corresponding to both His-pG and IgG, confirming successful conjugation of both biomolecules in SENDS (Fig. 2D). Additionally, SEM micrographs revealed an increase in Feret's diameter of SENDS compared to pristine ZIF-8 nanoparticles (Fig. 2B, Supplementary Fig. 1D), while AFM characterization showed increased surface roughness of SENDS (Supplementary Fig. 2). Phosphotungstic acid staining was used to aid in the visualization of the pG-IgG layer surrounding ZIF-8 under TEM, which was measured to be approximately 15 nm thick (Fig. 2B, Supplementary Fig. 3). The PXRD diffractogram of SENDS is highly similar to that of ZIF-8, suggesting that the functionalization process did not alter the crystal structure of ZIF-8 (Fig. 2C). The hydrodynamic diameter of SENDS was measured to be $416 \pm 10\,nm$, which was higher than pristine ZIF-8 diameter of $359 \pm 4\,nm$ (Supplementary Fig. 1E), indicating the presence of biomolecules on the nanoparticle surface. Additionally, the change in the surface zeta potential of ZIF-8 after His-pG functionalization, from $-23.2\,mV$ to $-17.0\,mV$, suggests the successful functionalization of the nanoparticle surface with His-pG, which is positively charged. Similarly, the zeta potential of the nanocomposites decreased to $-23.8\,mV$ after IgG conjugation, further confirming the attachment of negatively charged IgG (Fig. 2E). The amount of IgG bound to SENDS was quantified using the Bradford Coomassie Blue (BCA) assay, which showed a binding capacity of 29 mg of IgG per gram of ZIF-8 (Supplementary Fig. 1F).

To further investigate the His-pG-facilitated oriented assembly of IgG in SENDS, fragment-specific secondary antibodies were used to assess the relative availability of Fab to Fc regions on oriented IgG in SENDS (Supplementary Fig. 4A). ZIF-8 nanoparticles having non-specifically adsorbed IgG without the His-pG intermediate (ZIF-IgG) were used as the control. Fc-specific FITC and Fab-specific Cy3-labeled IgG were added in equal concentrations to ZIF-IgG and SENDS. We hypothesized that the oriented attachment of IgG on SENDS would enhance the accessibility of Fab sites and lead to increased binding of the Fab-specific Cy3-labeled IgG when compared to ZIF-IgG (Supplementary Fig. 4B). Upon washing and centrifugation to remove unbound secondary antibodies, the ratio of Cy3 (Fab-specific) to FITC (Fc-specific) fluorescence in SENDS was approximately 64% higher compared to that in ZIF-IgG (Fig. 2F), suggesting a greater availability of the target-recognizing Fab relative to Fc regions in SENDS. This demonstrates the important role of the His-pG linker in optimizing the orientation of IgG assembly in SENDS to improve accessibility of the stomata-targeting Fab region of IgG.

Following the successful synthesis of SENDS, its ability to retain biorecognition functionality after prolonged storage under ambient conditions was evaluated using the Fab-specific Cy3-labeled secondary antibody (Supplementary Fig. 4C). Cy3 fluorescence, which serves as an indicator for the availability of targeting sites on SENDS, remained stable over two weeks (Supplementary Fig. 4D). This suggests that SENDS can maintain their biorecognition functionality and stomata-targeting ability after extended storage under ambient conditions.

## SENDS target stomata on the leaf epidermis

We next sought to evaluate the stomata-targeting ability of SENDS on the leaf surface utilizing the biorecognition mechanism imparted by the LM6 IgG (Fig. 3A). The fluorescent dye fluorescein was encapsulated within SENDS to facilitate visualization of the nanoparticles under confocal laser scanning microscopy (CLSM). Fluorescein-encapsulating, unfunctionalized ZIF-8 was used as a control nanoparticle that lacks the stomata-targeting ability. Fluorescein was encapsulated through an in situ synthesis approach during MOF nanoparticle formation[49]. Spectroscopic characterization confirmed the encapsulation of fluorescein within ZIF-8 and SENDS, as evidenced by the appearance of a fluorescence peak at 520 nm corresponding to the fluorescein cargo (Supplementary Fig. 5C). Dye-encapsulating nanoparticles were washed with polyvinylpyrrolidone (PVP) to remove surface-adsorbed dyes, and the retention of the fluorescence peak at 520 nm after washing indicates that the fluorescein cargo was trapped within the MOF framework rather than being surface-adhered (Supplementary Fig. 5D). Additionally, the morphology, size and crystal structure of fluorescein-encapsulating nanoparticles were similar to the pristine ones, suggesting that dye-encapsulation did not significantly affect the physical properties of the nanoparticles (Supplementary Fig. 5A, E).

Dispersions of either fluorescein-encapsulating ZIF-8 or SENDS in MES buffer were then sprayed onto the abaxial epidermis, which contains abundant stomata, of *A. thaliana* leaves. This method of delivery mimics how a majority of nanofertilizers and nanopesticides are currently applied in the agricultural field. The leaf surface was then gently rinsed with deionized water before imaging. CLSM micrographs revealed the localization of SENDS along the guard cells of the stomata, whereas untargeted ZIF-8 nanoparticles appeared randomly distributed across the epidermis (Fig. 3C). By quantifying the proportion of fluorescent signals localized at the stomata from CLSM micrographs, it was determined that the colocalization rate of SENDS on stomatal guard cells was nearly 15-fold higher than that of plain ZIF-8, thus demonstrating the stomata-targeting efficacy of SENDS (Fig. 3B). To further assess the target recognition specificity of LM6 IgG towards stomata, ZIF-8 functionalized with a mouse monoclonal IgG (denoted ZIF-IgG(m)), which lack arabinan specificity, was also tested. CLSM

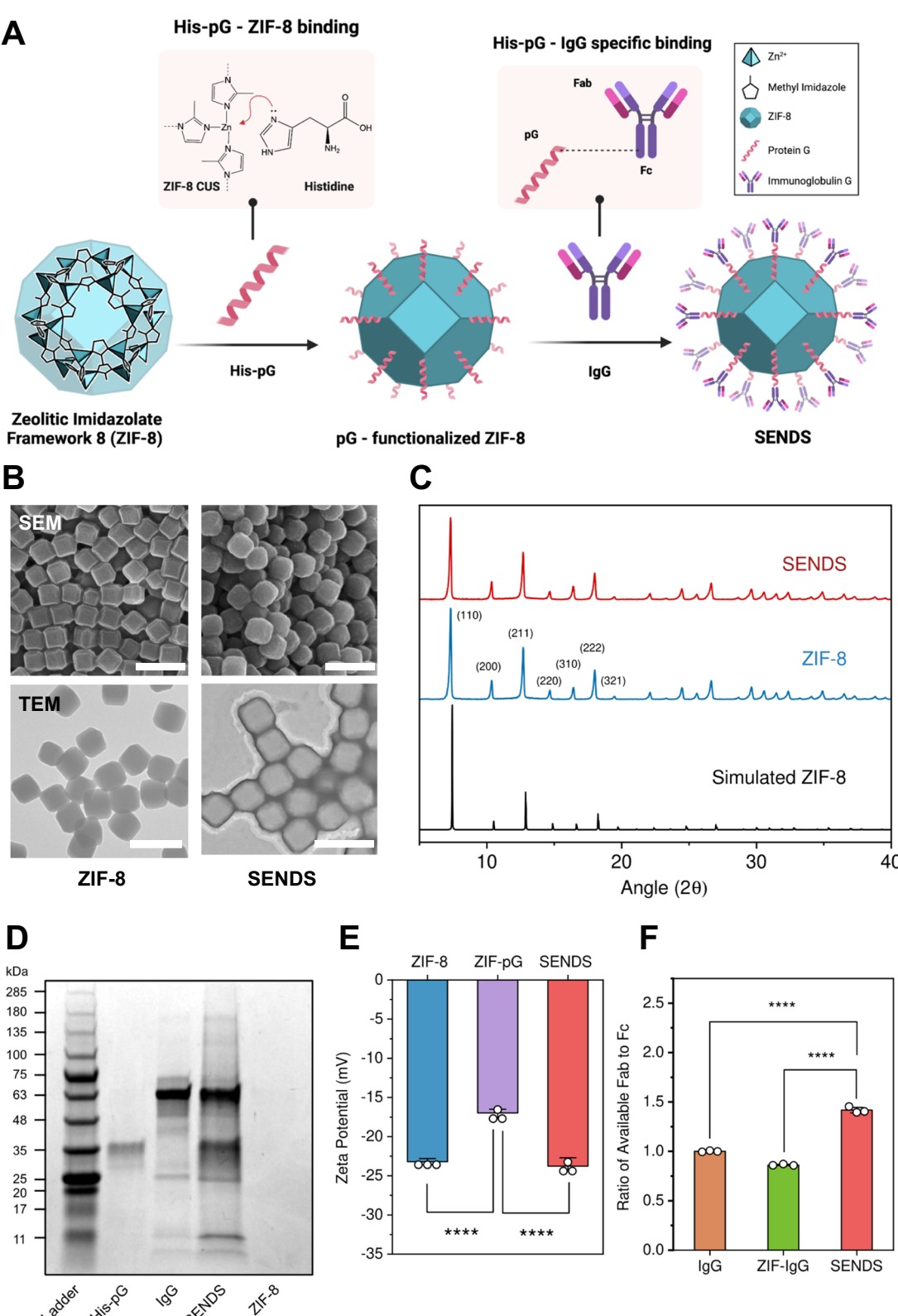

**Fig. 2 | Synthesis and characterization of SENDS. A** The assembly process of SENDS from individual components. Created in BioRender. Lew, T.T.S (2025) https://BioRender.com/2hzlzk6. **B** SEM (top) and TEM (bottom) micrographs of ZIF-8 (left) and SENDS (right). Scale bar, 500 nm. Imaging was repeated three times independently with similar results. **C** PXRD diffractogram of ZIF-8 and SENDS in comparison to a simulated ZIF-8 diffractogram. **D** SDS PAGE analysis.

**E** Zeta potential of ZIF-8, ZIF-pG and SENDS. **F** Ratio of available Fab to Fc regions for pure IgG, ZIF-IgG and SENDS based on ratio of Cy3 to FITC fluorescence. Data are presented as mean ± SD ($n = 3$ technical replicates). Statistical differences were calculated using one-way analysis of variance (ANOVA) with Tukey's post hoc test. ****$P < 0.0001$. Exact $p$ values are reported in the source data.

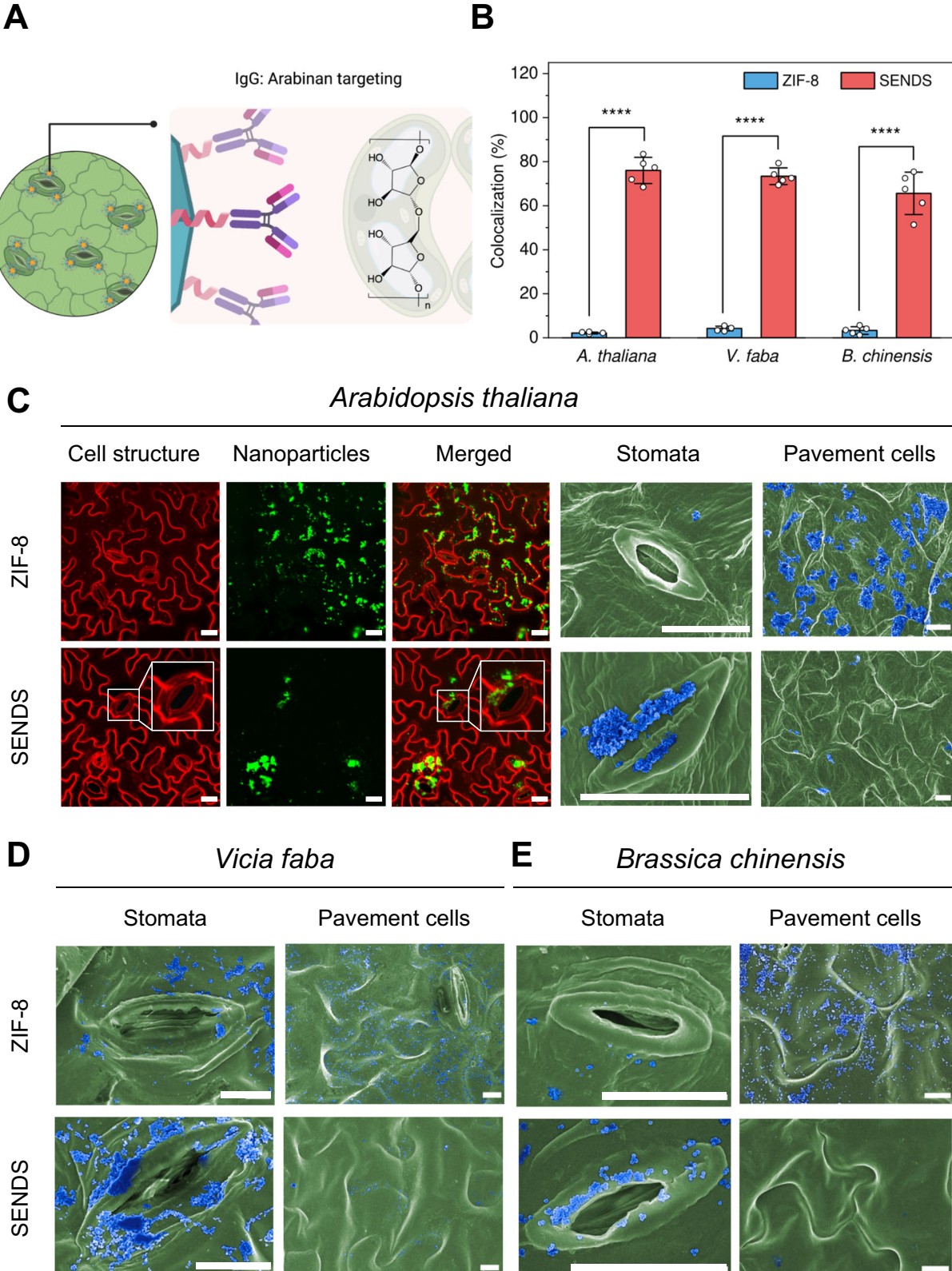

**Fig. 3 | Stomata localization of SENDS on the leaf epidermis. A** Targeting mechanism of SENDS. Created in BioRender. Lew, T.T.S. (2025) https://BioRender. com/wluiw9q. **B** Colocalization percentage of SENDS on *A. thaliana*, *V. faba* and *B. chinensis* based on CLSM micrographs. Data are presented as mean ± SD (*n* = 5 technical replicates). Statistical differences were calculated using two-sample t-test (two-sided). ****P < 0.0001. Exact *p*-values are reported in the source data. **C** CLSM (scale bar, 20 μm) and SEM micrographs (scale bar, 10 μm) showing the localization of ZIF-8 and SENDS on the leaf epidermis of *A. thaliana*. CLSM inset: magnified view of stomata and SENDS localization pattern. **D**, **E** SEM micrographs showing localization of plain ZIF-8 and SENDS (blue) on the leaf epidermis (green) of *V. faba* and *B. chinensis*. Scale bar, 10 μm. Relevant source data provided as a source data file.

colocalization results showed that ZIF-IgG(m) nanoparticles exhibited a random distribution on the leaf epidermis similar to untargeted ZIF-8, in contrast to the distinct guard cell localization observed with SENDS (Supplementary Fig. 6A).

In addition to CLSM, SEM was also performed as an orthogonal imaging technique to further validate the guard cell localization of SENDS. For SEM imaging, the same treatment procedure was used to introduce the nanoparticles to plants. The samples were then lyophilized and sputter-coated for imaging. Secondary electron imaging confirmed the localization of SENDS on the guard cells while non-targeted ZIF-8 was randomly distributed, consistent with the CLSM images (Fig. 3C). It is worth noting that the lyophilization process in preparation for SEM imaging could have compromised the structural integrity of the MOF nanoparticles, causing them to deform and aggregate to varying extents, leading to the loss in their original cubic shape. Nonetheless, SEM electron-dispersive spectroscopy (EDS) confirmed the identity of the nanoparticle aggregates deposited on or around the guard cells as SENDS (Supplementary Fig. 6B).

To demonstrate the applicability of our stomata-targeting SENDS across different plant species, we also studied the localization of SENDS on the leaf surface of two other dicotyledonous species – *B. chinensis* (pak choy), an economically important vegetable crop in Asia, and *V. faba* (fava bean), a model system in stomatal physiology research[50] – as well as two monocotyledonous species, *O. sativa* (rice) and *H. vulgare* (barley), both of which are important food crops. Micrographs obtained for all species exhibited similar localization patterns of SENDS on the guard cells of the stomata to that of *A. thaliana* (Fig. 3D, E, Supplementary Figs. 7, 8). Additionally, the higher colocalization percentage of SENDS compared to unfunctionalized ZIF-8 for all four plant species confirmed the stomata-targeting efficacy of SENDS across diverse plant species (Fig. 3B, Supplementary Fig. 9), indicating that targeting arabinan of guard cells can be a conserved strategy for the design of stomata-targeting nanocarriers for different plant species.

Beyond stomata localization, we next assessed how rainfall might impact nanoparticle adhesion, which is crucial for practical applications where external environmental factors may affect the performance of nanomaterials on the leaf surface. To study this, we evaluated the rainfastness of ZIF-8 and SENDS under two simulated rainfall conditions (2.5 mm and 5 mm), using methods previously reported in other literature[11,51,52] (Supplementary Fig. 10A). Results showed that under 2.5 mm rainfall, SENDS exhibited approximately 10% higher adhesion (86%) compared to ZIF-8 (75%) (Supplementary Fig. 10B). A potential explanation for this discrepancy could be the hydrogen bonding interactions between the biomolecules on the surface of SENDS and compounds in the leaf cuticle waxes[11,53,54], which may enhance the adhesion of SENDS on the leaf surface. However, at 5 mm rainfall, the adhesion of ZIF-8 and SENDS was similar, at approximately 60% to 70% (Supplementary Fig. 10C).

## In vitro antimicrobial activity of alkaloid-encapsulating ZIF-8

Stomata-targeting nanoparticles offer a strategic approach to bolstering plant defenses, as stomata serve as critical gateways for pathogen entry into host tissue. Building on the demonstrated stomata-targeting capability of SENDS, we next examined its efficacy in inhibiting bacterial colonization through in-vitro antimicrobial activity studies. Previous research has demonstrated that ZIF-8, the core framework of SENDS, displays antibacterial effects arising from the $Zn^{2+}$ ions it provides. These $Zn^{2+}$ ions act as antimicrobial agents by disrupting the structural integrity of bacterial cell walls, leading to bacterial death and growth inhibition[55,56]. In addition to its inherent antimicrobial properties, ZIF-8 has the advantage of a highly porous structure with numerous pore cavities that enable drug encapsulation - a feature widely exploited in the biomedical field[36,37,57]. In this work, we harnessed ZIF-8's encapsulating ability to host an antimicrobial cargo, thereby enhancing the antimicrobial efficiency of SENDS. We selected

sanguinarine chloride (SC), a plant-derived alkaloid, as the antimicrobial compound and encapsulated it in ZIF-8 via in situ synthesis, henceforth referred to as SC@ZIF. This selection is justified by sanguinarine's strong antimicrobial activity and its approval for use in plant-based livestock feed, ensuring its biosafety in agriculture and aquaculture[58–60]. The efficiency of SC encapsulation in ZIF-8 was quantified by measuring the absorbance of the supernatant from the SC@ZIF synthesis solution at the characteristic absorbance maximum of SC (475 nm) (Supplementary Fig. 11). At the highest SC loading concentration used for this work, encapsulation efficiency was approximately 90%.

To study the antibacterial activity of SENDS, *Xanthomonas campestris* was selected as the model phytopathogen as it is one of the most pervasive pathogens affecting cruciferous crops globally, causing black rot in economically significant crops such as cabbage, broccoli, kale, and pak choy[61,62]. To determine the optimal composition of SENDS for effectively inhibiting the proliferation and bacterial activity of *X. campestris*, colony forming unit (CFU) assays were conducted to assess the minimum bactericidal concentration (MBC) of pure ZIF-8, SC and SC@ZIF. Based on the results of the CFU assays, the MBC of pure ZIF-8 and SC were 75 µg/mL and 6 µg/mL respectively (Supplementary Fig. 12), while the MBC of SC encapsulated in 25 µg/mL ZIF-8 was 7 µg/mL (Figs. 4A, B, Supplementary Fig. 14A). This demonstrates that the encapsulation of SC in ZIF-8 enhanced its antimicrobial efficiency, as evidenced by the significant reduction in ZIF-8 concentration required to achieve complete pathogen suppression, from 75 µg/mL to 25 µg/mL. Additionally, SEM was used to investigate the effect of SC@ZIF on the pathogen (Fig. 4C), revealing morphological changes induced by the treatment. Untreated cells displayed a largely smooth and uniform surface, whereas cells treated with SC@ZIF at MBC exhibited pronounced wrinkling and shrinkage, suggesting disruption of the cell membrane caused by SC@ZIF.

## Stomata-targeted delivery of SENDS reduces pathogen internalization and virulence

Having demonstrated the in vitro antimicrobial activity of SC@ZIF, we next evaluated the combined effects of the stomata-targeting and antimicrobial properties of SC@SENDS through in vivo experiments. *B. chinensis* plants were treated with SC@SENDS or SC@ZIF suspended in MES buffer and left undisturbed for 24 h before inoculation with *X. campestris*. Leaf discs were cut, ground, and plated 24 h after pathogen introduction to quantify bacterial abundance of *X. campestris* in the apoplast of *B. chinensis* plants (Fig. 5A)[63]. Stomata-targeting SC@SENDS reduced the internalization of *X. campestris* (bacteria counts per mm² of leaf) by approximately 400-fold, 200-fold and 20-fold compared to untreated, SC and SC@ZIF-treated plants, respectively (Fig. 5B, Supplementary Figs. 13, 14B). This suggests that stomata-targeting SC@SENDS conferred orders of magnitude improvement in plant defense against pathogen invasion.

To confirm that the reduction in pathogen internalization was due to the stomata-targeting and antibacterial properties of SC@SENDS, rather than nanoparticle-induced stomatal closure, a separate experiment was conducted where plants were treated with SENDS and stomatal aperture was characterized under a light microscope (Supplementary Fig. 15). The average stomatal aperture in SENDS-treated leaves was similar to that in leaves treated with MES buffer, indicating that SENDS did not induce stomatal closure (Fig. 5C). A similar trend was observed in leaves treated with SC and SC@SENDS (Supplementary Fig. 15C), confirming that the observed reduction in pathogen internalization was due to the antibacterial properties and stomata-targeting capability of SC@SENDS. Additionally, when leaves were treated with SENDS combined with abscisic acid (ABA), a hormone known to induce stomatal closure, the response was comparable to treatment with pure ABA, further demonstrating that SENDS have minimal impact on stomatal aperture regulation. This underscores the

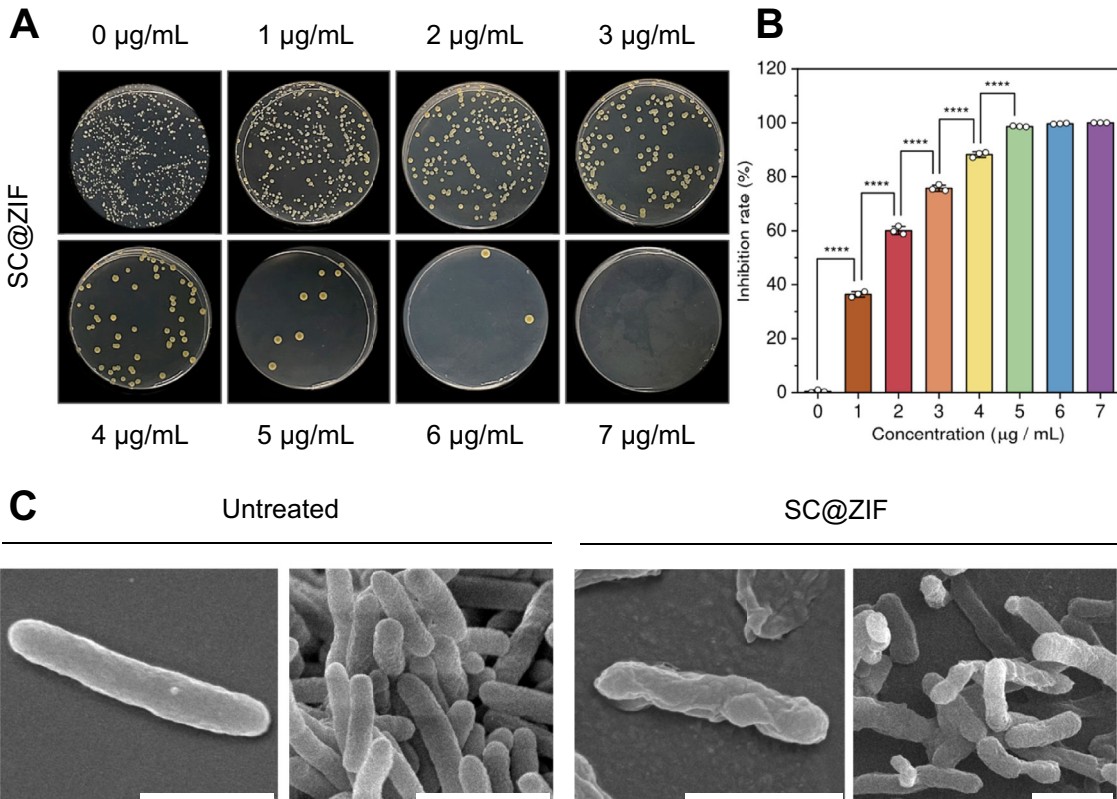

**Fig. 4 | In vitro antimicrobial activity of alkaloid-encapsulating ZIF-8. A** CFU assay for different concentrations of SC loaded into 25 μg/mL ZIF-8. **B** Inhibition rate of SC@ZIF with increasing concentration of SC loaded into 25 μg/mL ZIF-8. Data are presented as mean ± SD ($n = 3$ independent biological replicates). Statistical differences were calculated using one-way ANOVA with Tukey's post hoc test.

****$P < 0.0001$. Exact p-values are reported in the source data. **C** SEM micrographs showing morphological differences between treated and untreated *X. campestris*. Scale bar, 1 μm. Imaging was repeated three times independently with similar results. Relevant source data provided as a source data file.

critical role of SENDS's stomata-targeting capability in enhancing plant resistance to pathogen invasion without disrupting normal stomatal function. By directly addressing pathogen internalization at the stomata, key entry points to the apoplast, this targeted nanocarrier approach confers plants with a significantly more effective defense against bacterial colonization compared to non-targeted treatments.

Confocal microscopy, combined with the Live/Dead (SYTO 9/ propidium iodide) assay, was used to visualize the antimicrobial activity of SC@SENDS around the stomata (Fig. 5D). Untreated samples showed colonization of *X. campestris* around the stomata, with most bacteria appearing green, indicating viability. In contrast, SC@SENDS treated samples demonstrated a significant reduction in bacterial viability in the vicinity of the stomata, with dead bacteria appearing red (Fig. 5D). Quantification of live and dead bacteria by manual counting revealed up to a 70% reduction in bacterial viability around the stomata following SC@SENDS treatment (Supplementary Fig. 16B).

In addition to quantifying bacterial load in the apoplast, other in vivo tests were conducted to compare the effects of untargeted SC@ZIF and stomata-targeting SC@SENDS against *X. campestris* infection in terms of overall plant health, focusing on phenotypic differences and changes in photosynthetic efficiency. 10 days post-inoculation (10 dpi) of *X. campestris*, untreated and SC@ZIF-treated plants exhibited early symptoms of black rot caused by *X. campestris*, including characteristic wedge-shaped chlorotic lesions at the leaf margins and small dark spots on the abaxial side of the leaves (Fig. 5E, Supplementary Fig. 17A). In contrast, SC@SENDS-treated plants displayed no visible signs of infection, indicating near-total inhibition and effective prevention of further bacterial proliferation. Over the following 10 days, lesions on severely infected plants expanded towards

the midrib, causing wilting, leaf drop, and the development of darkened necrotic tissue by 20 dpi. While some chlorosis was observed in SC@SENDS-treated plants, this was likely due to natural leaf senescence rather than bacterial pathogenicity, as no symptoms of black rot were present (Supplementary Fig. 17D). To quantify the extent of infection, disease index (DI) scores were assigned to individual leaves of the inoculated plants based on an index system proposed by Guy et. al.[64] (Supplementary Fig. 17C). The overall DI for each plant was calculated by averaging the DI scores for all leaves. Over the 20-day test period, DI scores for untreated and SC@ZIF-treated plants increased significantly, while SC@SENDS-treated plants consistently maintained a DI score of 0, indicating no visible symptoms of black rot (Supplementary Fig. 17B).

Throughout the twenty-day period during which phenotypic differences were monitored, leaf samples were collected and chlorophyll-a (Chl-a) fluorescence measurements were performed to assess photosynthetic performance, a reliable indicator of overall plant health[65] (Supplementary Fig. 18). Three key parameters are of interest: Fv/Fm, representing the maximum photosystem II (PSII) quantum efficiency and photosynthetic capacity; ΦPSII, representing PSII quantum yield and the proportion of light absorbed used for photochemistry; and NPQ, which quantifies non-photochemical quenching, a mechanism by which plants dissipate excess light energy as heat to protect themselves against photodamage[66–68]. Figure 6A presents false-color maps illustrating the variation of Fv/Fm, ΦPSII and NPQ values across the leaf surface. In areas with lesions, all three parameters are low, indicating reduced PSII efficiency and a compromised ability to dissipate excess light energy, suggesting significant damage to the photosynthetic apparatus. Additionally, ΦPSII is consistently low across the entire surface of

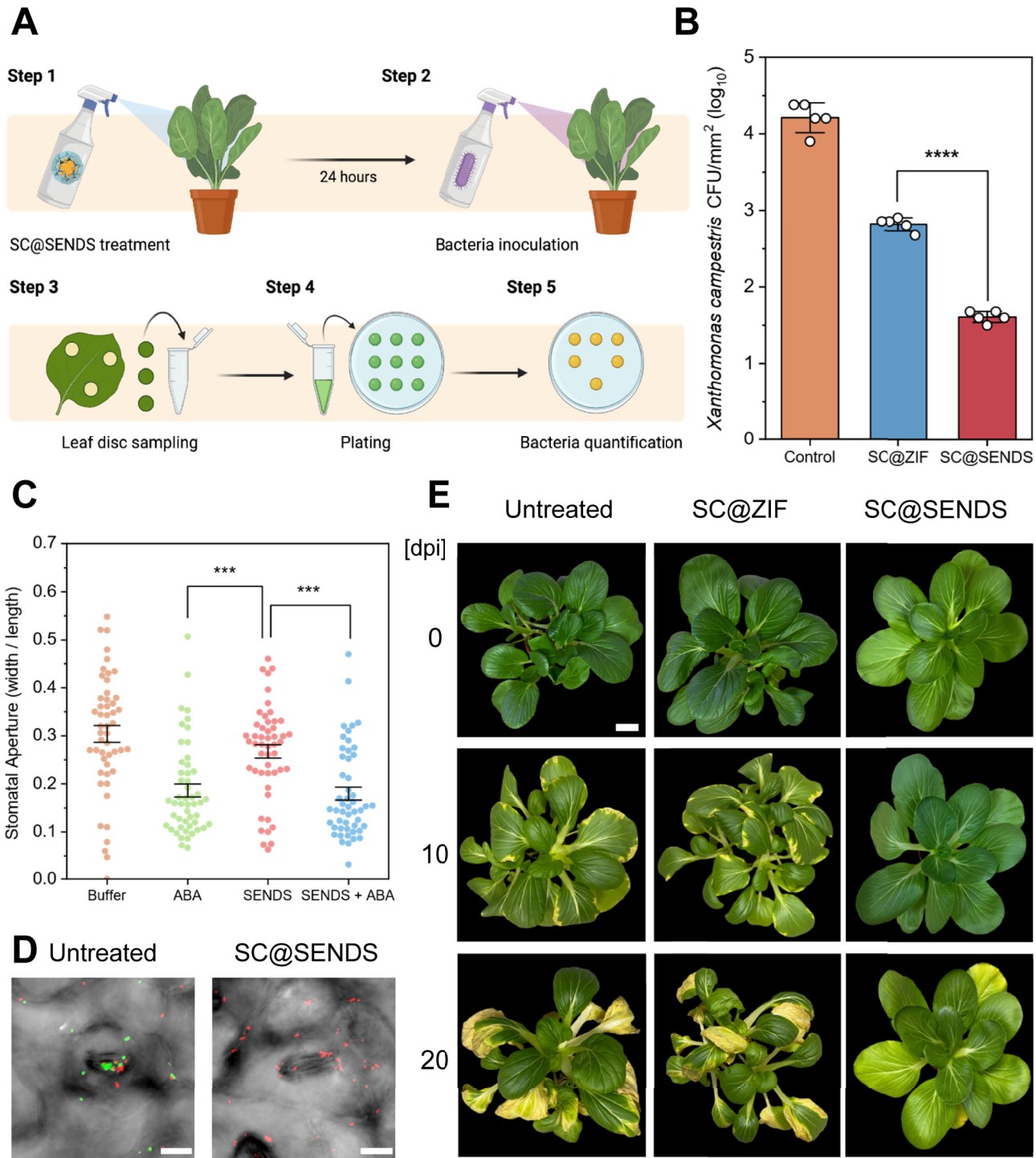

**Fig. 5 | In vivo antimicrobial activity of SC@SENDS. A** Schematic describing *X. campestris* quantification in the apoplast of *B. chinensis*. Created in BioRender. Lew, T.T.S. (2025) https://BioRender.com/bhsud4a. **B** Bacterial loads of *X. campestris* in leaf apoplast, quantified as CFU per mm², in *B. chinensis* plants (*n* = 5 independent biological replicates). Statistical differences were calculated using two-sample t-test (two-sided). ****P < 0.0001. **C** Stomatal aperture measurements (*n* = 50 technical replicates). Data are presented as mean ± SD. Statistical differences were calculated using one-way ANOVA with Tukey's post-hoc test. ***P < 0.001, ****P < 0.0001. Exact p-values are reported in the source data. **D** Confocal micrographs illustrating *X. campestris* entry to the stomata and the antimicrobial activity of SC@SENDS towards *X. campestris* surrounding the stomata. Scale bar, 10 μm. **E** Phenotypic differences between SC@SENDS, SC@ZIF and untreated *B. chinensis* infected with *X. campestris* over a period of 20 days. Scale bar, 3 cm. Relevant source data provided as a Source data file.

severely infected leaves from both untreated and SC@ZIF-treated plants, indicating that bacterial infection affects the energy conversion efficiency throughout the leaf. Low NPQ combined with low Fv/Fm and ΦPSII observed in untreated plants is likely a result of dysfunctional or damaged photosynthetic machinery. On another hand, the high NPQ in combination with low ΦPSII in SC@ZIF-treated plants may represent an overactive protective response mounted by the plant which reflects heightened stress[69]. In contrast, SC@SENDS-treated samples consistently display high Fv/Fm and ΦPSII values, alongside moderate NPQ, indicating that the leaves maintained their

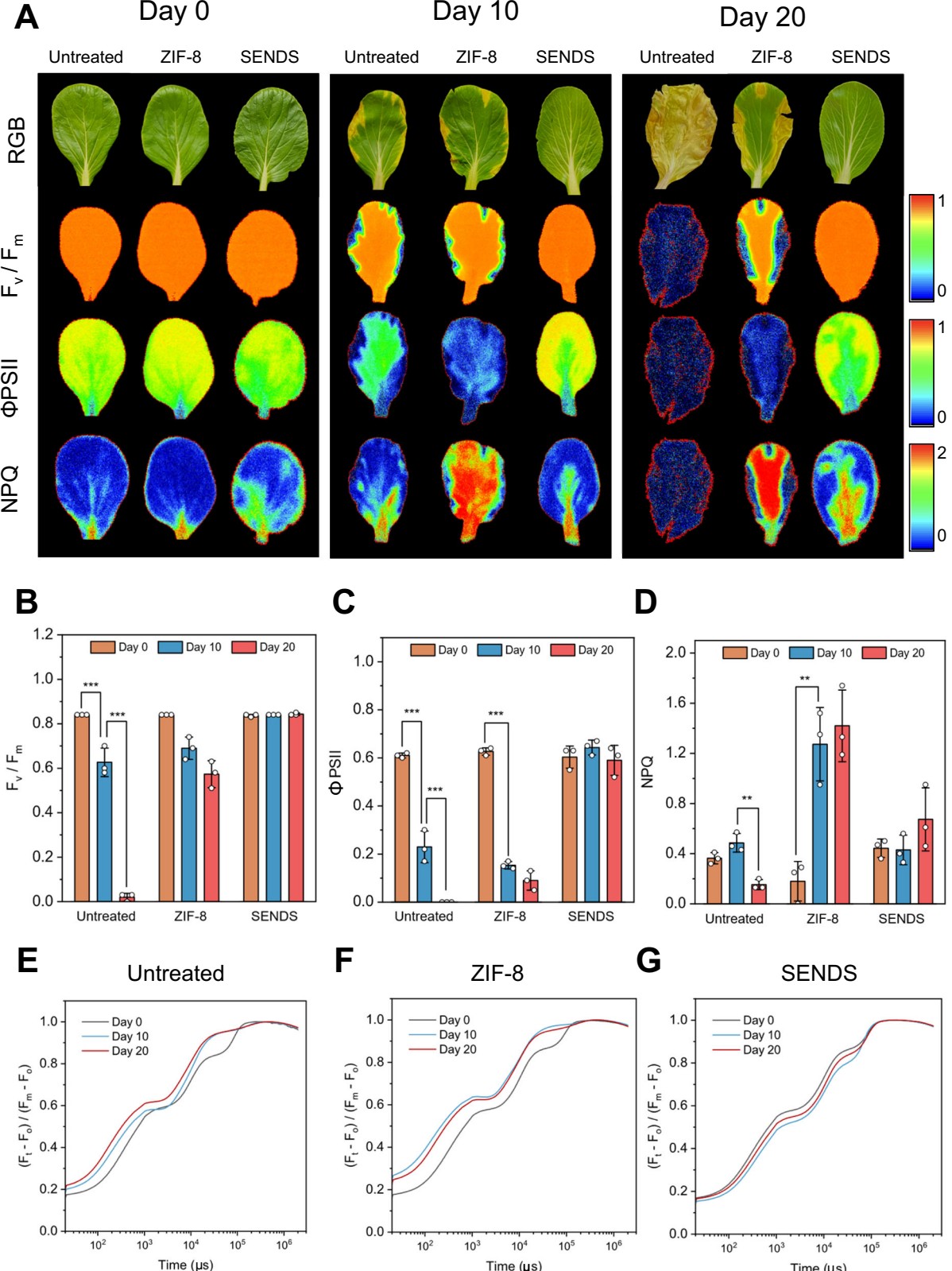

**Fig. 6 | Improved plant resistance against pathogen infection endowed by stomata-targeted nanoparticles. A** False-color maps of sampled leaves from *X. campestris*-infected plants illustrating the distribution of Fv/Fm, ΦPSII and NPQ values across the surface of the leaves. Plants were either untreated, treated with untargeted (SC@ZIF) or stomata-targeted nanoparticles (SC@SENDS) prior to pathogen challenge. Time-profile of (**B**) Fv/Fm, (**C**) ΦPSII and (**D**) NPQ of sampled leaves across the twenty-day experimental duration. Data are presented as mean ± SD (*n* = 3 independent biological replicates). Statistical differences were calculated using two-sample t-test (two-sided). **$P < 0.01$, ***$P < 0.001$. Exact p-values are reported in the source data. Evolution of OJIP transient in (**E**) untreated, (**F**) SC@ZIF and (**G**) SC@SENDS-treated samples. Relevant source data provided as a source data file.

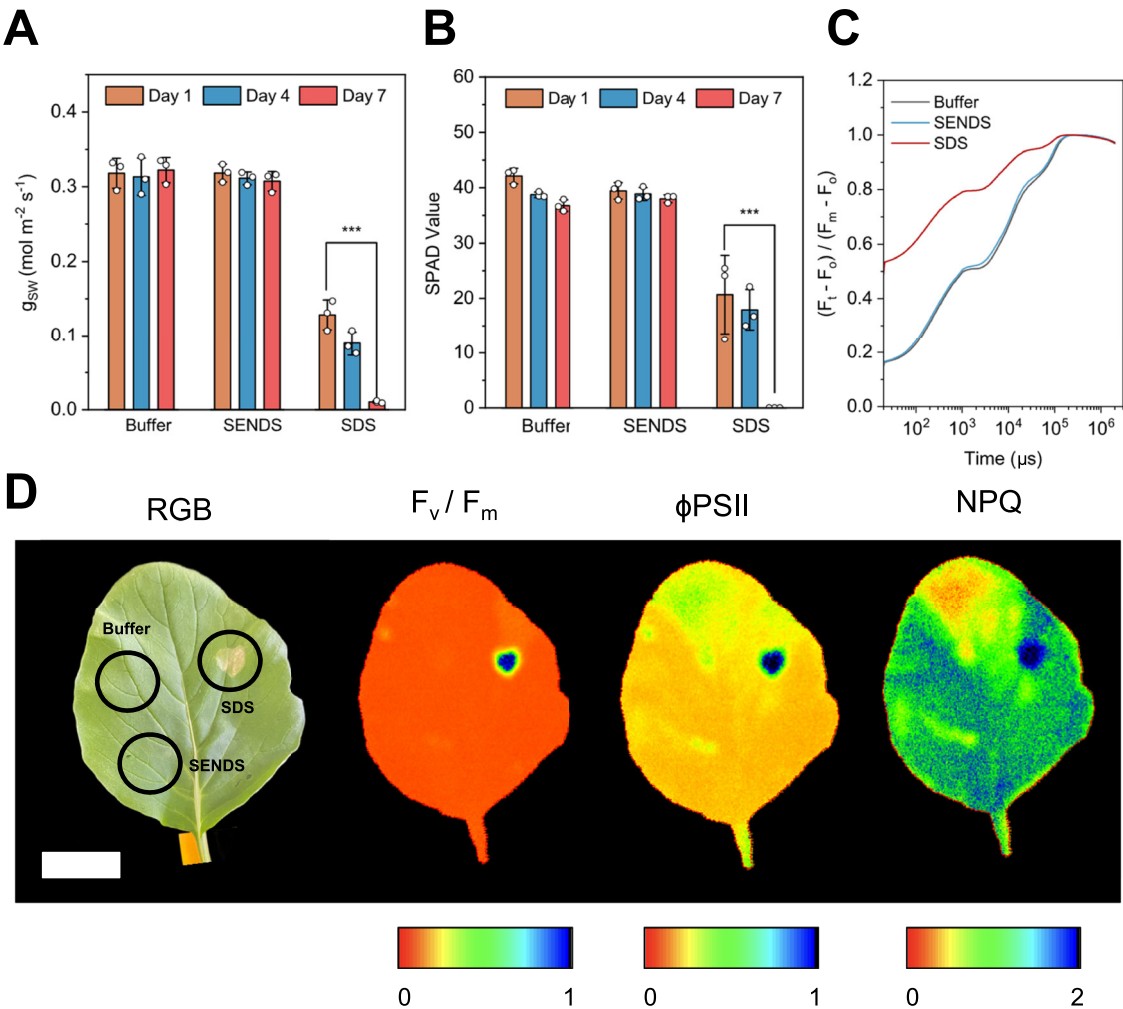

**Fig. 7 | Biocompatibility studies of SENDS in plants.** Time-profile of (**A**) stomatal conductance, (**B**) SPAD and (**C**) OJIP transients of leaves treated with buffer, SENDS and SDS over a week. Data are presented as mean ± SD ($n = 3$ independent biological replicates). Statistical differences were calculated using two-sample t-test (two-sided). ***$P < 0.001$. Exact p-values are reported in the source data. **D** False-color map of a leaf treated in several localized areas on the seventh day post-treatment. Scale bar, 3 cm. Relevant source data provided as a source data file.

physiological health despite pathogen challenge. These results further suggest that SC@SENDS conferred living plants with improved resistance against invading pathogens owing to its stomata-targeting property, as compared to non-targeted SC@ZIF.

The effects of bacterial pathogenicity are also evident upon analyzing the evolution of the 3 photosynthetic parameters in the different treatment groups over the duration of the tests (Fig. 6B–D). Sharp decreases in Fv/Fm and ΦPSII were observed in the severely infected untreated and SC@ZIF-treated samples, while these values remained consistently high for SC@SENDS-treated plants (Fig. 6B, C). Additionally, fluctuations in NPQ observed in untreated and SC@ZIF-treated samples, in conjunction with their lowered Fv/Fm and ΦPSII values, suggest damage to photosynthetic machinery and heightened stress. In contrast, the consistently moderate NPQ in SC@SENDS-treated plants further indicates that no significant stress symptoms developed during the experiment (Fig. 6D). Furthermore, we analyzed the integrity of the electron transport chain in the plant's photosynthetic apparatus using the OJIP transient curve, a fluorescence induction curve with distinct O, J, I, and P phases. Each of these phases reflect key steps in the photosynthetic electron transport process and is widely used to evaluate overall photosynthetic performance[70,71]. In healthy plants, the phases O, J, I, and P are clearly distinguishable (Supplementary Fig. 18C), as demonstrated by the fluorescence curves for SC@SENDS-treated samples (Fig. 6G). In contrast, measurements

taken for untreated and SC@ZIF-treated samples show that the transition between the I and P phases becomes less distinct at 10 and 20 dpi (Fig. 6E, F). This blending of the I and P phases indicates impaired electron transport, suggesting that the PSII reaction centers are damaged as a result of high stress or bacterial infection.

## Biocompatibility of SENDS in living plants

To assess the biosafety of SENDS, we conducted a comparative study involving three groups: (a) a negative control treated with MES buffer, (b) SENDS suspended in MES buffer, and (c) sodium dodecyl sulfate (SDS) as a positive control. SDS was selected as a positive control for its known phytotoxic properties by disrupting cellular membranes and inhibiting growth[72]. Each treatment was applied as a foliar spray to the abaxial side of *B. chinensis* leaves, and the plants were monitored over a week to observe for any adverse effects. In addition to monitoring Chl-a fluorescence profile, stomatal conductance and chlorophyll content were assessed.

Stomatal conductance ($g_{SW}$) measures the rate of gas exchange between the plant and the atmosphere through the stomata. Under optimal conditions, stomata open to facilitate the entry of $CO_2$ for photosynthesis and the exit of water vapor for transpiration[73,74]. Conversely, when the plant is stressed or if stomatal machinery is damaged, $g_{SW}$ may decrease below optimal values, which generally range from 0.2 to 0.5 mol m$^{-2}$ s$^{-1}$ for healthy *B. chinensis*. Our results

show that SDS treatment caused $g_{SW}$ to fall below 0.15 mol m$^{-2}$ s$^{-1}$. In contrast, leaves treated with SENDS maintained healthy $g_{SW}$ values around 0.3 mol m$^{-2}$ s$^{-1}$ throughout the tests (Fig. 7A), indicating that the SENDS did not impair the stomatal activity of living plants over time. Chlorophyll content, which is measured in SPAD values, serves as an indicator of plant health and nitrogen status. High SPAD values indicate higher chlorophyll concentration and enhanced photosynthetic capacity, while lower values may signal stress or nutrient deficiency[75,76]. SDS treatment resulted in significantly lower SPAD values, while SENDS treatment showed no significant difference compared to the negative control (Fig. 7B). This finding is supported by Chl-a fluorescence measurements, which revealed a decrease in Fv/Fm and ΦPSII in SDS-treated leaves. Additionally, NPQ values were initially high, indicating significant stress, and decreased over time due to the increasing area of necrotic tissue (Supplementary Figs. 19, 20). In contrast, SENDS treatment maintained consistently high Fv/Fm and ΦPSII values, with moderate NPQ throughout the experiment, suggesting no immediate signs of stress.

The OJIP transient profile on the seventh day post-treatment confirmed that, while compromised PSII electron transport was evident in the SDS-treated samples, no such impairment was observed in the SENDS-treated plants (Fig. 7C). Furthermore, localized application of each treatment on the abaxial side of leaves and subsequent Chl-a fluorescence assessment on the seventh day showed that SDS resulted in necrotic tissue formation with low Fv/Fm, ΦPSII, and NPQ at the application sites. In contrast, SENDS-treated sites showed similar photosynthetic capabilities and morphology of the leaves as buffer-treated sites (Fig. 7D). Overall, these results suggest that SENDS are biocompatible, with no adverse effects on leaf physiology or morphology.

## Discussion

In this study, we introduced a stomata-targeting nanocarrier system, SENDS, and demonstrated its application to enhance disease resistance in plants. SENDS consist of a porous ZIF-8 nanoparticle core, designed for high loading of antibacterial cargo, and are decorated with protein ligands to enable biorecognition-mediated targeting of stomata guard cells. Both the nanoparticle core and surface ligands can be synthesized and functionalized through a straightforward room temperature-process, providing an accessible and efficient method for producing SENDS. By including a polyHis6-pG moiety in the biomolecule assembly, we improved the accessibility of the stomata-targeting Fab regions of LM6 IgG, enhancing SENDS targeting efficiency. Stomata localization studies demonstrated the targeting effect of SENDS, achieving up to 15-fold higher stomata localization compared to non-targeted nanoparticles. This enhanced targeting efficiency was consistent across multiple dicotyledonous and monocotyledonous species, including *A. thaliana*, *B. chinensis*, *V. faba*, *O. sativa* and *H. vulgare*, highlighting the broad applicability of this nanocarrier approach.

Furthermore, by encapsulating a natural plant-derived alkaloid within the framework of SENDS, we demonstrated through in vivo studies that the combined stomata-targeting effect and antimicrobial action of encapsulated cargo effectively reduced pathogenic internalization and virulence of *X. campestris* in *B. chinensis* plants by 20-fold compared to untargeted nanocarriers. While *X. campestris* may also enter through the hydathodes to initiate virulence[77], our results demonstrate that stomata-targeting nanoparticles could provide an effective strategy to enhance plant resistance to bacterial invasion. By precisely targeting the stomatal guard cells, this nanotechnology-based approach offers enhanced protection against pathogen internalization, without altering the natural stomatal movement in response to environmental cues. This nanocarrier-mediated targeted disruption of pathogen invasion addresses a critical vulnerability in plant defense systems. Notably, SENDS can be applied via a simple foliar spray, similarly to conventional pesticides, eliminating the need for specialized infrastructure or application methods.

In addition to their efficacy, we assessed the biosafety of the nanoparticles by quantifying the stomatal conductance, photosynthetic efficiency and chlorophyll concentration in nanoparticle-treated plants. No adverse effects on stomatal conductance or induction of stress responses were observed, confirming that the stomata-targeting nanoparticles exhibit good biocompatibility in living plants. These outcomes demonstrate the safety of the nanoparticles for agricultural applications, offering a potent yet non-invasive method for enhancing plant tolerance towards biotic stresses. In the future, the porous framework of SENDS can be leveraged to encapsulate and deliver other cargoes, including pesticidal compounds, antimicrobial peptides, and double-stranded RNAs (dsRNA), which show great promise for crop protection against pathogens[78]. While SENDS demonstrate significantly enhanced effectiveness over non-targeted nanocarriers, future studies should evaluate their potential to reduce agrochemical use and associated costs while maintaining efficacy. The reliance of SENDS on antibody-based targeting may also pose cost challenges for large-scale production. Exploring the efficiency of alternative stomata-targeting ligands, such as small nanobodies or non-protein-based moieties, could further improve scalability and affordability. Coupled with advancements in large-scale MOF production, the SENDS platform presented here can guide the design of more cost-effective stomata-targeting nanocarriers for agricultural applications[79]. Additionally, future studies should also evaluate these nanocarriers in real-world field conditions, assessing their efficacy and environmental impact to ensure their effectiveness and safety in large-scale deployment.

Overall, this study advances plant nanotechnology by developing a rationally designed MOF-based nanocarrier platform for stomata-targeted delivery, demonstrating its potential to bolster plant defense against pathogen colonization. It offers an alternative or complimentary approach to genetic engineering for improving plant stress tolerance, with potential applicability to a broad range of plant species. This could pave the way for more precise and effective nanotechnology-based strategies to enhance plant disease resistance, accelerating progress in advancing agricultural productivity and sustainability.

## Methods

### Preparation of ZIF-8 and its guest-encapsulating variants

Two aqueous stock solutions were prepared for ZIF-8 synthesis: 5 mL of 2.72 M 2-methylimidazole (2 MIM) with 0.55 mM cetyltrimethylammonium bromide (CTAB), and 5 mL of 0.270 M zinc acetate dihydrate. After sonication to dissolve the precursors, the solutions were mixed and vigorously swirled for 30 s, causing the mixture to rapidly turn cloudy. The mixture was left undisturbed for 2 h, then centrifuged at 9000 $g$ for 10 min. The resulting precipitate was washed thrice with methanol and dried overnight under vacuum at 60 °C. Guest molecule encapsulation was performed in-situ by adding the compounds to the Zn$^{2+}$ precursor and stirring for 5 min before mixing with the 2-MIM precursor. For F@ZIF, 0.5 mL of 5 mM fluorescein in methanol was added for a 10 mL ZIF-8 synthesis system. For SC@ZIF, different concentrations of aqueous sanguinarine chloride were used.

### Preparation of SENDS and ZIF-IgG nanoparticles

Histidine-tagged protein G was purchased from Genscript (Z02007), and the Anti-Pectic Polysaccharide (alpha-1,5-arabinan) LM6 IgG Antibody was purchased from Kerafast (ELD008). 2 mg/mL protein G was prepared by dissolving the lyophilized product in ultrapure water. For bioconjugation, 4 mg of dry ZIF-8 powder was dispersed in 2 mL of 0.05% Tween-20 solution, and 125 μL of the prepared protein G solution was added. The mixture was stirred overnight at 150 rpm. Afterward, the solution was centrifuged at 9000 $g$ for 3 min, and the supernatant was replaced with fresh 0.05% Tween-20. Then, 100 μL of the LM6 IgG solution was added and the mixture

was stirred overnight at 150 rpm. Finally, the solution was centrifuged again at 9000 g for 3 min, and the SENDS were dispersed in 10 mM MES at pH 7.5. Preparation of SC or fluorescein-encapsulating SENDS follows the same procedure using SC@ZIF or F@ZIF in place of ZIF-8.

To prepare ZIF-IgG(m) for specificity studies, pG-functionalized F@ZIF was dispersed in 0.05% Tween-20. His-tagged HRP mouse monoclonal IgG was purchased from Santa Cruz Biotechnology. 60 μL of the 200 μg/mL solution was added and the mixture was stirred overnight at 150 rpm. Washing and storage procedures follow that of SENDS.

## Materials characterization

SEM micrographs were captured using the JSM-7610F Plus FESEM (JEOL Ltd.) and EDS was performed using the X-Max 80 electron dispersive spectrometer by Oxford Instruments. Samples were sputter-coated with platinum prior to imaging. TEM imaging was performed using JEM-2100F FETEM (JEOL Ltd.). 1% phosphotungstic acid at pH 7 was used for staining to aid in visualization of the pG-IgG layer. PXRD measurements were performed using a Bruker D8 ENDEAVOR diffractometer. Zeta potential and hydrodynamic size was measured using the Zetasizer Nano (Malvern Instruments). Measurements were conducted at 25 °C and pH 7.5, with the Smoluchowski method applied to calculate Zeta potential. Fluorescence or absorbance measurements for F@ZIF-8, BCA assay and antibody orientation experiments were performed using the Biotek Synergy H1 plate reader by Agilent Technologies.

For SDS-PAGE analysis to verify protein compositions on SENDS, the samples were first boiled in Laemmli buffer at 95 °C for 5 min. Afterwards, they were loaded into a 4−20% gradient gel and electrophoresis was carried out at 100 V for 120 min in in the presence of 1X Tris-glycine buffer with 0.1% (w/v) SDS. Coomassie Blue was used to stain the gel, which was imaged using the Azure 300Q Chemiluminescent Imaging System.

## Antibody orientation experiments

Fab fragment specific Cy3-conjugated AffiniPure Goat Anti-Rat IgG (Item No. 112-165-006) and Fc fragment specific Fluorescein (FITC)-conjugated AffiniPure Goat Anti-Rat IgG (Item No. 112-095-008) were purchased from Jackson ImmunoResearch Pte Ltd. First, 30 μL each of 0.20 mg/mL Cy3 and FITC IgG was added to 340 μL of (1) deionized water (control), (2) a pure LM6-IgG sample prepared by mixing 85 μL of LM6 IgG solution with 255 μL deionized water, (3) 2 mg/mL ZIF-IgG and (4) 2 mg/mL SENDS in deionized water. The samples were vortexed to ensure thorough mixing and incubated at 37 °C for 3 h. After incubation, the samples were centrifuged at 9000 g for 3 min and the supernatant containing unbound antibodies was replaced with fresh deionized water. Finally, the samples were transferred to a black 96-well plate and fluorescence measurements were performed. The excitation and emission wavelengths used were 495 nm/520 nm for FITC and 540 nm/570 nm for Cy3.

## Stability tests

SENDS (2 mg/mL) were stored in deionized water in a glass vial under ambient conditions. At 0, 7, and 14 days, 340 μL aliquots were extracted for testing. Each aliquot was mixed with 30 μL of 0.20 mg/mL Cy3-labeled AffiniPure Goat Anti-Rat IgG, vortexed, and incubated at 37 °C for 3 h. Following incubation, the samples were centrifuged at 9000 g for 3 min and the supernatant containing unbound antibodies was replaced with fresh deionized water. Finally, the samples were transferred to a black 96-well plate and fluorescence measurements were performed (Ex: 540 nm/Em: 570 nm).

## Plant growth conditions

*A. thaliana, B. chinensis, V. faba* and *H. vulgare plants* were grown in soil for 3 to 4 weeks in a plant growth chamber (Aralab FITOCLIMA 600 PLH) under a 14-h light, 10-h dark photoperiod, 50% relative humidity, and a constant temperature of 23 °C. *O. sativa* was grown in Kimura B nutrient solution under a 16-h light, 8-h dark photoperiod, with 28 °C/25 °C day/night temperature and 50%/70% day/night humidity. For confocal imaging, ML1pro:mCherry-RCI2A *A. thaliana*[80] seedlings were grown on Murashige and Skoog agar (½ strength) for 14 days.

## Nanoparticle localization and imaging experiments

F@ZIF-8 and F@SENDS (2 mg/mL) suspended in 10 mM, pH 7.5 MES buffer (5 mL) were sprayed onto the abaxial side of the leaves of all five plant species. The plants were left undisturbed for 30 min, after which the treated leaves were rinsed with deionized (DI) water and 5 × 5 mm samples were cut for microscopy. For confocal microscopy (Olympus FV3000), leaf samples were mounted on glass slides. *B. chinensis, V. faba, O. sativa* and *H. vulgare* samples were stained with propidium iodide (20 μg/mL). Colocalization efficiency of nanoparticles was analyzed using Fiji by ImageJ. An intensity threshold was applied to the GFP channel and colocalized fluorescent signals (fluorescein) were quantified as a percentage of total fluorescent signals in a single field of view (Eq. 1). The total area assesse for each leaf sample was approximately 0.225 mm². For SEM, the plants were treated with ZIF-8 and SENDS in place of their fluorescein-encapsulating variants. The leaf samples were immersed in liquid nitrogen for 2 min and subsequently freeze-dried overnight before imaging.

$$\text{Colocalization (\%)} = \frac{\text{Total colocalized fluorescence intensity}}{\text{Total fluorescence intensity}} \times 100\%$$

(1)

## Rainfastness test

*B. chinensis* leaves of similar leaf areas were separately treated with 10 μL F@ZIF-8 and F@SENDS prepared at a concentration of 2 mg/mL in deionized water. The treated leaves were left to dry for 3 h to ensure deposition of the materials onto the leaf surface. Once dried, each leaf was mounted on a glass slide and positioned at a 60-degree angle using a retort stand. The surface of the leaves was then rinsed with diH₂O, with the volume of simulated rain determined based on rainfall intensity and leaf surface area (e.g., 2.5 mm rainfall for 5 cm² leaf area corresponds to 1.25 mL of simulated rain). The wash liquid was collected in separate vials for each treatment, and the amount of material washed off was quantified by measuring the fluorescence for F@ZIF-8 and F@SENDS (Ex: 495 nm, Em: 520 nm). The material retention (%) of each treatment was determined by comparing the measured values in the wash liquid to those of the stock solutions diluted by an equivalent volume of wash liquid (Supplementary Fig. 10).

## Quantification of alkaloid encapsulation efficiency

A calibration curve for SC was constructed by preparing aqueous solutions of the compound ranging from 0.25 to 2.5 mM. Absorbance of each solution was measured at 475 nm. To quantify encapsulated SC, the supernatant from SC@ZIF synthesis was collected and absorbance was measured to determine amount of unencapsulated SC. The amount of unencapsulated SC was subtracted from the known initial concentration of SC to calculate the encapsulated SC. To confirm the encapsulated SC value, complete acid degradation of the SC@ZIF was performed using hydrochloric acid. The absorbance of the resulting solution was then measured to provide complimentary confirmation of the encapsulated SC content.

## In vitro antibacterial activity of nanoparticles

Agar plates for in-vitro antibacterial studies were prepared using an autoclaved 1877 ISP medium composed of 5 g tryptone, 3 g yeast extract, 15 g agar, and 1 L deionized (DI) water. *X. campestris* (ATCC

33913) was inoculated from a frozen stock culture and incubated on the 1877 ISP agar plates at 25 °C for three days. For the antibacterial studies, *X. campestris* colonies were resuspended in 300 μL of sterile DI water and diluted from $10^8$ CFU/mL (-0.25 OD) to $10^4$ CFU/mL. Subsequently, different dilutions of ZIF-8, SC or SC@ZIF suspended in 30 μL of sterile DI water were combined with 270 μL of $10^4$ CFU/mL *X. campestris* solution in a well plate. A negative control containing none of the antibacterial agents was included. The samples were incubated overnight at 25 °C and plated onto the prepared agar plates on the next day. The plates were incubated for three days at 25 °C and resulting colonies were enumerated by manual counting.

To investigate the effect of SC@ZIF on *X. campestris*, the bacteria were mixed with an aqueous SC@ZIF suspension for 3 h. After incubation, the cells were fixed overnight in a 1:1 solution of 8% glutaraldehyde and paraformaldehyde. Following fixation, the cells were dehydrated through a graded ethanol series and subsequently lyophilized for SEM imaging.

### Pathogen infection and bacterial load quantification

SC@ZIF and SC@SENDS (3 mg/mL) suspended in 10 mM, pH 7.5 MES buffer (5 mL) were sprayed onto the abaxial side of *B. chinensis* leaves and left undisturbed for 24 h. The plants were then infected by spraying the abaxial side with $10^8$ CFU/mL *X. campestris* in a 5 mL 0.05% Silwet-77 solution. After inoculation, the plants were placed overnight in an enclosed chamber with ~80% relative humidity. For SC controls, the leaves were treated with 5 mL of SC at a concentration of 1 mg/mL in deionized water in place of the nanoparticles.

For quantification of bacterial load in the apoplast of the plants, mature, fully expanded leaves from treated plants of the same growth stage were selected by numbering leaves and chosen using a random number generator. The leaves were gently wiped with ethanol-soaked wipes to remove surface contamination. Using a 4 mm circular cutter, 4 leaf discs were excised ~1 cm from the midrib, avoiding large veins and the leaf edges. Two leaf discs from each side of the midrib were sampled, yielding a total of 4 leaf discs, each with an area of 12.6 mm². The leaf discs were placed into a 2 mL Eppendorf tube containing 200 μL of sterile DI water and thoroughly crushed using the back end of a plate spreader, constituting one replicate. Five replicates were conducted for each treatment group. Following the crushing of the samples, the supernatant was carefully removed, and a ten-fold serial dilution was performed across seven dilutions. Utilizing the drop plate assay, 5 μL of the original or diluted samples was applied to 1877 ISP agar plates, which were then incubated at 25 °C for three days. After incubation, colonies were enumerated by manual counting, and the number of colony-forming units (CFUs) of *X. campestris* per mm² of leaf area was calculated and recorded.

For phenotyping tests, plants were treated and left under a 14-h light, 10-h dark photoperiod with 50% relative humidity, and a constant temperature of 23 °C in the plant chamber. Photographs were taken at 10 and 20 dpi. The overall plant disease index for a single plant was calculated by averaging the DI scores for all its leaves. Replicates were taken from plants of the same growth stage. Chl-a measurements were conducted using the Fluorcam 1300 (Photon Systems Instruments) during the same light/dark photoperiod each day to ensure consistency. Mature, fully expanded leaves from treated plants of the same growth stage were selected by numbering leaves and chosen using a random number generator. The leaves were then cut and mounted on the sample platform with double-sided tape. After 20 min of dark adaptation, the predefined protocol 'Light Curve 1' (Supplementary Fig. 11A, B) was used to perform Chl-a fluorescence measurements. OJIP transient measurements were performed using the Fluorpen FP100 (Photon Systems Instruments). Leaf clips provided with the device were used to dark-adapt localized regions of leaves from the treated plants for

20 min. Finally, OJIP was measured using the predefined protocol for OJIP measurement.

### Visualizing the antimicrobial activity of SC@SENDS

SC@ZIF and SC@SENDS (3 mg/mL) suspended in 10 mM, pH 7.5 MES buffer (5 mL) were sprayed onto the abaxial side of *B. chinensis* leaves and left undisturbed for 24 h. The LIVE/DEAD Baclight bacterial viability kit by Invitrogen (L7012) was used to carry out fluorescent labeling of *X. campestris* for confocal microscopy. Prior to inoculation, 5 mL of $10^8$ CFU/mL *X. campestris* was stained using only SYTO 9 from the kit, following the fluorescence microscopy protocol provided by the manufacturer. The stained bacteria were then sprayed onto the abaxial side of the leaves. After 6 h, 5 × 5 mm samples were excised, stained with only propidium iodide from the kit (prepared according to manufacturer's instructions) for 2 min and were subsequently mounted on glass slides for confocal microscopy. Bacterial viability was determined by enumerating the live and dead bacteria surrounding the stomata (Eq. 2). The total area analyzed for each leaf sample was approximately 0.020 mm².

$$\text{Bacterial viability}(\%) = \frac{\text{Number of live bacteria (green)}}{\text{Number of dead bacteria (red)}} \times 100\% \quad (2)$$

### Stomatal aperture measurements

*B. chinensis* leaves were treated with a stomatal opening buffer (50 mM KCl + 10 mM MES/KOH) and left under white light at 22 °C for 1h. Subsequently, the leaves were treated with either MES buffer (10 mM, pH 7.5), SENDS, SC@SENDS (3 mg/mL), SC (1 mg/mL), ABA (10 μM final concentration) or SENDS in ABA and incubated for another 1 h under the aforementioned conditions. Following this, the leaf samples were excised and mounted with the abaxial epidermis facing upwards for microscopy using the Axio Imager M2 upright microscope (Zeiss). Stomatal apertures were quantified as the ratio between the width and length of the stomata, which was determined using Fiji by ImageJ (Supplementary Fig. 15).

### Biocompatibility of nanoparticles in plants

*B. chinensis* plants were treated either MES buffer (10 mM, pH 7.5), SENDS or SDS (5%) and left under a 14-h light, 10-h dark photoperiod with 50% relative humidity, and a constant temperature of 23 °C in the plant chamber. Measurements were taken on the first, fourth- and seventh-days post-treatment. Chl-a and OJIP measurements were taken using the protocols described above. Stomatal conductance was measured using the LI-600 Porometer (LI-COR) and chlorophyll content was measured using the CHL PLUS chlorophyll meter (atLEAF). CHL values were converted to SPAD values[76].

### Statistics and reproducibility

All data are presented as mean ± standard deviation, unless otherwise noted. Student's t-test (two-sided) was used to analyze data between two groups. Origin 2021b was used for statistical analyses. For data involving three or more groups, one-way analysis of variance (ANOVA) was conducted, followed by Tukey's multiple comparison test. Statistical significance was defined as ns ($P > 0.05$), *($P < 0.05$), **($P < 0.01$), ***($P < 0.001$), or ****($P < 0.0001$), with a 95% confidence interval.

### Reporting summary

Further information on research design is available in the Nature Portfolio Reporting Summary linked to this article.

## Data availability

All data supporting the findings in the paper are present in the main manuscript and the Supplementary Information. Source data is

available for Figs. 2–7 and supplementary Figs. 1, 2, 4, 5, 9–17, 19 and 20 in the associated source data file. Source data are provided with this paper.

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

## Acknowledgements

This work was supported by the Singapore National Research Foundation (NRF) under the NRF Fellowship (Award No: NRF-NRFF15-2023-0002) and by the Research Center for Sustainable Urban Farming, NUS (Grant No: A-8000149-00-00). X.Y. and O.S.L. acknowledge the support from the Singapore Ministry of Education through the Academic Research Fund (AcRF) Tier 2 (Grant No: MOE2019-T2-2-128). Schematic images were constructed using BioRender (https://www.biorender. com/). We thank Dr. Di Shen for discussions on pathogen-related experiments.

## Author contributions

S.P. and T.T.S.L. conceived the idea and designed the experiments. S.P., C.H.L., and C.S. developed the methodology, performed the experiments and analyzed the data. X.Y. assisted with stomatal aperture measurements and growing transgenic plants for localization experiments. O.S.L. and T.T.S.L. supervised the project and assisted with data interpretation. S.P., O.S.L., and T.T.S.L. co-wrote the manuscript. All authors discussed and analyzed the results.

## Competing interests

The authors declare no competing interests.
