## [Transparent Peer Review file · Nature Communications]

Stomata-Targeted Nanocarriers Enhance Plant Defense Against Pathogen Colonization

Corresponding Author: Dr Tedrick Thomas Salim Lew

Version 0:

Reviewer comments:

Reviewer #1

(Remarks to the Author)

Puangpathumanond et al. demonstrated the interactions between plants and nanoparticles by developing nanocarrier systems that specifically target stomata. This study offers valuable insights into designing nanomaterials aimed at enhancing crop protection. It is well-structured and promising, illustrating how a nanotechnology approach can improve plant disease resistance and contribute to sustainable agriculture.

1. Line 117: The author mentioned, "ZIF-8 was chosen because Zn²⁺ is an essential micronutrient for plants." Is the author intending to enhance the supply of Zn²⁺ using SENDS? If that is the case, this should be clearly demonstrated. If not, the manuscript needs to provide a more detailed or different explanation.

2. In line 165, the author indicates that the SEM micrographs showed a subtle increase in surface roughness of SENDS. However, the AFM analysis would provide a more accurate characterization of nanoscale surface roughness.

3. After applying SEND to crops, it may offer protection against pathogen infection for a certain duration. However, how stable is SEND, particularly the protein used to functionalize it? Can it be stored under ambient conditions, similar to other agrochemicals?

4. In the in vivo antimicrobial activity test, is there data demonstrating the effect of SC alone, without encapsulation? This should be added to highlight the effect of SENDS.

5. Since SENDS is a nanocarrier system designed for targeted delivery, it is recommended to quantify the nanoparticles after they are sprayed on the leaves. This quantification should focus on areas near the stomata, pavement cells, and mesophylls, as well as any nanoparticles that may be lost, such as those that fall into the soil.

6. In line 330, the author noted that some chlorosis observed in SC@SENDS-treated plants was likely a result of natural leaf senescence. If that is the case, healthy control plants should be included in Figure 5E for comparison.

7. The author chose 5% SDS as a positive control in the biocompatibility test. However, SDS is too harmful for plants, even at low concentrations. Using 5% SDS is excessive, as it causes immediate and noticeable damage to the leaves. It would be better to select a milder positive control. Additionally, conducting tests with various concentrations of SDS, especially at higher levels, would provide more valuable information.

8. The manuscript should include a discussion of the feasibility of applying this technology in practice, considering factors such as cost, large-scale production, and environmental impacts.

9. How long do SENDS remain attached to the leaf surface and the guard cells of the stomata? The imaging analysis was performed after a gentle rinsing. How strongly does SENDS bind to the guard cells of the stomata? Factors such as rain and wind conditions should be considered in field applications.

10. SENDS demonstrated 20 times greater effectiveness than a non-targeted nanocarrier in reducing the colonization of Xanthomonas. The implications of this finding should be discussed in more detail. How can this advantage of

nanotechnology be leveraged in agriculture? Is a 20-fold increase in effectiveness sufficient? A 20-fold increase in effectiveness at a certain concentration does not necessarily imply that a 20-fold lower concentration of the active reagent would yield the same result. It's important to note that achieving this level of targeting requires additional modifications and functionalization, which involve expensive proteins. It is recommended to include a discussion on how the 20-fold higher effectiveness can still be beneficial.

Minor comments

- Phosphotungstic acid staining was mentioned in the text. Indicate only Figures S1 and S2, but Figure 2 B should also be indicated. It is recommended to give a separate alphabet number to each SEM and TEM image.
- Figure 2C shows that stimulated ZIF-8 is not necessary.
- In lines 139, 148: A histidine-tagged protein might mean polyhistidine-tagged (typically His6) proteins. In the manuscript, it is written as if one histidine can form a coordination bonding with transition metal ions. This should be clarified.
- In line 170, the text mentions the hydrodynamic diameter of SENDS, but no data is shown. Data should be added.
- In line 177, the author mentions the BCA assay in the text at line 177, but the data are not shown or indicated.
- The experiment for the orientation of IgG is not easy to understand. The text indicates Fig. S3, but more explanation is needed in the main text to help readers understand.
- Enlarging a portion of the confocal micrographs of SENDS in Figure 3C can be confusing and may not be necessary. If this presentation approach is to be used, a description should be included in the figure caption.

Reviewer #2

(Remarks to the Author)

Summary

The manuscript presents an antimicrobial loaded MOF, decorated with functional antibody LM6 to target the stomata of crops and prevent bacterial infection. This work leverages advances in the biomedical field, material science, and plant bionanotechnology to create a sound and working material that populates the stomata after foliar spray and reduces the disease index of bacterial infection. Such work is considered to be a significant improvement over conventional agrochemicals and is a subject of interest to experts in many fields of study. The manuscript is well written, using proper English, and contains compelling figures, which will please the readers of the journal.

Although this work presents a good advancement in the field, several improvements need to be addressed before the publication of this manuscript.

Introduction

Although the introduction is well written and contains relevant references, there is no mention of the extensive use of antibodies, such as LM6, for immunohistological studies of plant cell walls. These have been utilized for over 15 years to target the delivery of gold nanoparticles to visualize specific tissues under electron scanning microscopy. The evidence of these biorecognition molecules for such studies is robust and not exclusive to the work of nano-enabled agriculture. Such work should be included and properly cited somewhere in lines 59-68.

Results

Some results are written in the present tense instead of the past tense. E.g. Line 272. "...while the MIC of SC encapsulated in 25 $\mu\text{g/mL}$ ZIF-8 is 7 $\mu\text{g/mL}$ "

By definition, the minimum inhibitory concentration (MIC) is the minimum concentration of an active required to inhibit the growth of a bacteria, based on the experiment this should be the concentration that prevents the bacteria from growing a CFU/mL higher than the initial inoculum. I believe that what is being described in the results is the minimum bactericidal concentration (MBC), which is the lowest concentration necessary to kill the bacteria, as evidenced by the CFU assay.

Line 330: "While some chlorosis was observed in SC@SENDS-treated plants, this was likely due to natural leaf senescence rather than bacterial pathogenicity, as no symptoms of black rot were present." This is observed in Fig 5 E. Although it might be reasonable to assume this, the chlorosis is not visible in Fig 6. (biocompatibility) or recorded to affect the photosynthetics of the leaves. Therefore, this chlorosis is most likely due to bacterial inoculation. To conclusively determine the cause of the chlorosis, additional observation time (dpi 25 or 30), as well as an untreated and non-inoculated control are needed.

Line 316: "This underscores the critical role of SENDS's stomata-targeting capability in enhancing plant resistance to pathogen invasion without disrupting normal stomatal function." Although SC@SENDS significantly improved the disease index, the effect of sanguinarine on stomatal opening was not assessed. Therefore, it is possible that SC@SENDS is causing stomatal closure and consequently, reducing bacterial colonization in the stomata. A SC@SENDS and sanguinarine control need to be included in the stomatal aperture experiment.

Additionally, proper controls should be included in the in vivo antimicrobial efficacy assays. These include an untreated and uninoculated control, as well as a sanguinarine treated plant group. Moreover, a commercial product based on plant alkaloids such as a sanguinarine-based pesticide should also be used as a comparison to contextualize the advancement of the SENDS technology.

Although SC@sanguinarine reduced the disease index no micrographic evidence was recorded of the material interaction directly with the bacterial. SEM or confocal microscopy should be employed to visualize how SENDS is preventing bacteria from colonizing the stomata. Appropriate controls should be assessed too, like untreated controls to visualize infection.

Discussion

Line 443: "By precisely targeting the stomatal guard cells, the primary entry points for pathogens into the apoplast, we demonstrated a nanotechnology-based approach to confer plants with enhanced protection against pathogen internalization..." This is not factual. For example, infections by *Xanthomonas campestris* (the pathogen used) on leaves have been shown to start at the hydathodes, at the edge of the leaf. Please see the following:

<https://doi.org/10.1016/j.cub.2023.01.013> and <https://doi.org/10.1104/pp.16.01852>

Furthermore, this is backed by symptoms used to establish the disease index in this work, as DI 1 and 2 mainly present chlorosis at the edges of the leaf. Considering this, it may well be that SC@SENDS protects the hydathodes rather than the stomata. LM6 antibody does not target specifically guard cells, but moieties in pectin that are present in several leaf tissues, including hydathodes.

Line 456: "Overall, this study presents a breakthrough in plant nanotechnology by introducing a rationally designed nanocarrier platform that precisely targets stomata to bolster plant defenses against pathogen colonization." This approach has been reported before (as referenced in the manuscript) and therefore does not qualify as a breakthrough. Additionally, gold nanoparticles tagged with these antibodies have been significantly reported for immunolabeling studies and visualization of rhamnogalacturonans for over 15 years. A more modest description of the work is suggested, as the study is a methodological improvement of previous studies using MOFs.

Although outside the scope of the study, there is no mention of the scalability or feasibility of using antibodies for agricultural purposes. It is appropriate to include one or two lines for non-experts to understand that such refined architecture and biotargeting motifs are not yet practical for mass production.

Materials and Methods

Several details regarding the nanoparticle localization and imaging experiments need to be specified. This includes: The equation utilized to quantify the material's colocalization of the stomata, the area (mm²) assessed on the leaves, the volume/concentration of SEND and Zif utilized.

Additionally, it is unclear why the treatments are rinsed before assessing the colonization. This leads to the removal of the loosely adhered material, leaving only the particles that are adhered to the stomata due to the antibody. In turn, this can skew the quantification of the colonization of material.

Several details are noticed in the antibacterial in vitro studies that can significantly affect the results. The recommended media for the bacteria is YGC medium, which was not utilized. This can lead to a reduced count of colonies forming on the agar. Furthermore, the initial inoculum of bacteria is relatively low at 10⁴ CFU/mL. It is recommended to perform the antimicrobial assessment at a bacterial load of 10⁵ - 10⁶ CFU/mL. This can also lead to a lower colony count. Finally, the antimicrobial treatments and the bacteria are suspended in DI water, which in itself is not conducive to bacterial growth. This can also lead to a lower colony count. It is recommended to perform this assay using YGC broth.

For the pathogen infection and bacterial load quantification experiments, the concentration of SC@SENDS is not described and the volume of both SENDS and bacteria is not specified. Furthermore, ethanol was used to rinse the leaves which can penetrate apoplast and reduce bacteria CFU. The use of wipes soaked in waterborne antiseptics, such as bleach, is recommended given that water is too viscous to readily penetrate the apoplast.

There are no criteria described for sampling the leaf tissue. This can lead to skewed results as tissue in the edge of the leaves can have higher inoculum compared to tissue in the center. Considering this, the number of samples for the bacteria apoplast bacteria quantification is rather small with n=3. There is no description of whether different leaves were sampled or a singular leaf at different areas. If few leaves are sampled, this can also skew the results, as healthy/sick leaves can be preferentially sampled. The experiments should also include an untreated and uninoculated control, as well as a sanguinarine treated plant group.

For phenotyping tests, no uninoculated control was included. Similarly, 3 leaves are not enough to evaluate an entire plant, unless specific criteria are set for sampling (such as sampling leaves from the same plant growth stage).

Reviewer #3

(Remarks to the Author)

General comments- This paper presents a rationally designed nanocarrier platform named surface ligand-engineered nanoparticles for targeted delivery to stomata (SENDS) for targeted delivery of antibiotics to stomata of a few common crops to enhance plant defense against pathogens. A stomata targeting antibody IgG was attached to the surface of ZIF-8 MOF core through modular assembly. The antibody functionalized nanocarriers have exhibited 15 times higher colocalization with stomata compared to unmodified MOF. The SENDS was loaded with a common antibiotic sanguinarine chloride (SC) and the SC loaded SENDS have exhibited antimicrobial activity both in vitro and in vivo, with 20 folds higher antimicrobial activity compared to the untargeted ZIF8, highlighting the improved efficacy in plant pathogen control by this targeted delivery strategy. Deployment of this material to stomata did not disturb stomata movements and plant photosynthetic activity. Although the concept of stomata targeting nanoparticles was previously reported, this is the first demonstration of improved antimicrobial efficacy achieved by targeted antibiotic delivery to stomata. The paper is very well written, topic of this paper is timely, and the proposed stomata targeting technology can be a promising and viable solution for plant pathogen control, provided that their scalability and non-toxicity are proven. To increase the impact of the study and to enhance the scientificity of the proposed work, I would recommend addressing the following:

1. Agriculture in general has a very thin profit margin, therefore agrochemicals need to be very affordable. The MOF and stomata targeting antibody can be very expensive to manufacture and the strategy may be not viable economically. The authors should discuss potential challenges and opportunities for implementing this technology at an agriculture-relevant scale and how to perform future studies that may support the feasibility of the proposed delivery platform.
2. The stomata targeting capacity of this material is demonstrated only on three types of dicot plants. Monocot plants, such as maize, wheat and rice also need to be studied to examine the universality and versatility of this technology.

3. Only the untreated plants are used as control for the in-vivo antimicrobial activity study. Another control with free SC at the same concentration is needed to examine the efficacy of this technology compared to conventional application methods.
4. A rain fastness study for SENDS ZIF-8 is necessary to examine the persistence and efficacy of this technology under more relevant condition.
5. The authors have used CTAB to adjust the shape of ZIF-8 and claimed the change in ZIF-8 morphology from dodecahedral to a cubic shape can improve the oriented assembly of the stomata-targeting biomolecules, hence stomata targeting efficacy of the material. However, CTAB is toxic and can pose risks to ecological and human health. A more compelling justification could be made by comparing the performance of the dodecahedral and cubic-shaped materials, which would strengthen the argument and highlight the advantages of the chosen geometry.
6. Fig. 1 lacks any data and is just a schematic. Perhaps it should be removed or integrated with Fig. 2.

Reviewer #4

(Remarks to the Author)

Version 1:

Reviewer comments:

Reviewer #1

(Remarks to the Author)

All inquiries and concerns have been addressed and resolved.

Reviewer #2

(Remarks to the Author)

The authors reviewed and edited the manuscript titled "Stomata-Targeted Nanocarriers Enhance Plant Defense Against Pathogen Colonization" in response to the reviewers' comments.

The manuscript was edited to include more background literature on biorecognition motifs for stomata targeting, added explanation on the scalability and future studies, and implemented the new results appropriately into the document. Supplementary rainfastness, antimicrobial, phytotoxicity, and bacterial localization experiments were completed by the authors in response to the reviewers' comments. The results from the new assays answer the comments made previously by the reviewers and are presented factually in the manuscript. The reviewer commends the authors for the additional experiments performed in response to the comments.

There are no additional comments on the manuscript, and it is recommended for publication in its current state.

Reviewer #3

(Remarks to the Author)

The authors have addressed the referee's comments well.

Reviewer #4

(Remarks to the Author)

Manuscript title: “Stomata-Targeted Nanocarriers Enhance Plant Defense Against Pathogen Colonization”

[Authors]: We would like to thank all reviewers for their constructive feedback on our manuscript. In this document, we have provided a point-by-point response to each reviewer’s comments in blue, and the changes made in the manuscript are highlighted in yellow.

REVIEWER COMMENTS

Reviewer #1 (Remarks to the Author):

Puangpathumanond et al. demonstrated the interactions between plants and nanoparticles by developing nanocarrier systems that specifically target stomata. This study offers valuable insights into designing nanomaterials aimed at enhancing crop protection. It is well-structured and promising, illustrating how a nanotechnology approach can improve plant disease resistance and contribute to sustainable agriculture.

[Authors]: We thank the reviewer for the supportive feedback on the significance and novelty of our work, and its potential as a new promising approach to improve plant disease resistance.

1. Line 117: The author mentioned, "ZIF-8 was chosen because Zn²⁺ is an essential micronutrient for plants." Is the author intending to enhance the supply of Zn²⁺ using SENDS? If that is the case, this should be clearly demonstrated. If not, the manuscript needs to provide a more detailed or different explanation.

[Authors]: We appreciate the reviewer’s feedback and have revised the manuscript to clarify that Zn²⁺ as a plant micronutrient is a possible secondary benefit, rather than a primary reason for selecting ZIF-8 as our nanocarrier of choice:

Line 121-126: “Specifically, zeolitic imidazolate framework 8 (ZIF-8), composed of tetrahedrally-coordinated Zn²⁺ ions and 2-methylimidazole, was selected for its proven ability to encapsulate and release various guest molecules^{31–35}, making it well suited for stomata-targeted agrochemical delivery. Additionally, Zn²⁺ is biocompatible and generally benign to plants, with some studies suggesting secondary benefits as a micronutrient^{29–31}.”

2. In line 165, the author indicates that the SEM micrographs showed a subtle increase in surface roughness of SENDS. However, the AFM analysis would provide a more accurate characterization of nanoscale surface roughness.

[Authors]: We agree with the reviewer that AFM characterization would provide a more rigorous substantiation of our claim regarding the surface roughness of SENDS. We have performed the AFM characterization which we included in the revised manuscript as **Figure S2**, with the corresponding discussion on the surface roughness of SENDS in **Lines 171-173**:

Lines 171-173: “Additionally, SEM micrographs revealed an increase in Feret’s diameter of SENDS compared to pristine ZIF-8 nanoparticles (Fig. 2B, S1) while AFM characterization showed increased surface roughness of SENDS (Fig. S2).”

Fig. S2. Surface roughness analysis of ZIF-8 and SENDS. AFM images and corresponding line profiles of (A) ZIF-8 and (B) SENDS demonstrate increased surface roughness following functionalization.

3. After applying SEND to crops, it may offer protection against pathogen infection for a certain duration. However, how stable is SEND, particularly the protein used to functionalize it? Can it be stored under ambient conditions, similar to other agrochemicals?

[Authors]: We agree with the reviewer that it is important to evaluate the stability of SENDS, particularly their targeting capabilities, for practical applications. To address this, we performed additional experiments to assess the biorecognition functionality of SENDS stored at room temperature for two weeks. At each time point – freshly synthesized, 1 week, and 2 weeks - we extracted aliquots of SENDS and added a Cy3-labelled secondary antibody that selectively binds to the arabinan-targeting Fab region of LM6 IgG on SENDS. With this approach, measured Cy3 fluorescence would serve as an indicator of available targeting sites on the nanoparticle surface and, therefore, the biorecognition functionality of SENDS. After incubation and washing to remove unbound antibodies, we measured Cy3 fluorescence and compared the values across the time points. Our results indicate similar Cy3 fluorescence from secondary antibodies introduced to SENDS at different time points over 2 weeks, suggesting that SENDS retains its targeting capabilities after 2-week storage under ambient conditions. The following data has been included as part of **Figure S4** in the revised manuscript:

Fig. S4. (C) Schematic detailing stability study protocol. **(D)** Minimal changes in Cy3 secondary antibody fluorescence over time suggest that the biorecognition function of the LM6 IgG adhered to SENDS remains intact during 2-week storage under ambient conditions.

Lines 199-204: “Following the successful synthesis of SENDS, its ability to retain biorecognition functionality after prolonged storage under ambient conditions was evaluated using the Fab-specific Cy3-labeled secondary antibody (Fig. S4C). Cy3 fluorescence, which serves as an indicator for the availability of targeting sites on SENDS, remained stable over two weeks (Fig. S4D). This suggests that SENDS can maintain their biorecognition functionality and stomata-targeting ability after extended storage under ambient conditions.”

4. In the in vivo antimicrobial activity test, is there data demonstrating the effect of SC alone, without encapsulation? This should be added to highlight the effect of SENDS.

[Authors]: We fully agree with the reviewer that the effect of SC alone needs to be investigated as a control group to highlight the importance of stomata-targeting SENDS. We conducted additional in vivo experiments to quantify the apoplastic bacterial load in plants treated with unencapsulated SC at the same concentration as in SC@SENDs. Our results showed that free SC did not significantly reduce bacterial load, whereas SC@SENDs achieved a 2.5 to 3 order of magnitude greater reduction compared to free SC, further highlighting the benefits of encapsulation and stomata-targeted delivery provided by SENDS. We have added this additional data to **Figure S13**:

Fig. S13. (C) Drop plate assay for uninoculated control showing no CFU growth. **(D)** Drop plate assay for free SC-treated *B. chinensis* with 10^8 CFU/mL *X. campestris*. **(E)** Average bacterial load in leaf apoplast of free SC-treated plants in comparison to untreated and SC@SENDS treated plants. Error bars indicate s.d. (n=5). Statistical differences were calculated using two-sample t-test. ***P<0.001, ****P<0.0001.

Lines 322-325: “Stomata-targeting SC@SENDS reduced the internalization of *X. campestris* (bacteria counts per mm² of leaf) by approximately 400-fold, 200-fold and 20-fold compared to untreated, SC and SC@ZIF-treated plants, respectively (Fig. 5B, S13, S14B).”

5. Since SENDS is a nanocarrier system designed for targeted delivery, it is recommended to quantify the nanoparticles after they are sprayed on the leaves. This quantification should focus on areas near the stomata, pavement cells, and mesophylls, as well as any nanoparticles that may be lost, such as those that fall into the soil.

[Authors]: We thank the reviewer for the feedback and agree that the potential loss of SENDS to the soil is an important consideration for assessing its suitability under practical situations. To assess the adhesion of SENDS under precipitation, we conducted rainfastness tests¹⁻³ on ZIF-8 and SENDS under two simulated rainfall intensities: 2.5 mm and 5 mm. Our results indicate that under 2.5 mm rainfall, SENDS demonstrated roughly 10% higher adhesion compared to ZIF-8 (86% vs 75%). This discrepancy could potentially be due to hydrogen bonding interactions between the biomolecules on the SENDS' surface and compounds in the cuticular waxes of the leaf epidermis^{1,4,5}. At 5 mm rainfall, however, ZIF-8 and SENDS exhibited similar adhesion, suggesting higher rainfall intensities may overcome the initial adhesion advantage of SENDS. Furthermore, quantification of SENDS distribution on the leaf surface has previously been addressed in Figure 3B, where we used confocal micrographs to visualize and quantify their localization between stomata and pavement cells.

The experimental protocol and new results of our rainfastness studies have been included in the revised manuscript in **Lines 265-274** and as new **Figure S10**.

Lines 265-274: “Beyond stomata localization, we next assessed how rainfall might impact nanoparticle adhesion, which is crucial for practical applications where external environmental factors may affect the performance of nanomaterials on the leaf surface. To study this, we evaluated the rainfastness of ZIF-8 and SENDS under two simulated rainfall conditions (2.5 mm and 5 mm), using methods previously reported in other literature^{11,50,51} (Fig. S10A). Results showed that under 2.5 mm rainfall, SENDS exhibited approximately 10% higher adhesion (86%) compared to ZIF-8 (75%) (Fig. S10B). A potential explanation for this discrepancy could be the hydrogen bonding interactions between the biomolecules on the surface of SENDS and compounds in the leaf cuticle waxes^{11,52,53}, which may enhance the adhesion of SENDS on the leaf surface. However, at 5 mm rainfall, the adhesion of ZIF-8 and SENDS was similar, at approximately 60% to 70% (Fig. S10C).”

Fig. S10. Rainfastness studies. (A) Schematic detailing rainfastness experimental protocol. Rainfastness of SC, ZIF-8 and SENDS under (B) 2.5 mm and (C) 5 mm rainfall.

6. In line 330, the author noted that some chlorosis observed in SC@SENDS-treated plants was likely a result of natural leaf senescence. If that is the case, healthy control plants should be included in Figure 5E for comparison.

[Authors]: We thank the reviewer for this suggestion and have added the phenotypic progression of an uninoculated plant over 25 days to **Fig. S17**, showing signs of chlorosis from senescence of older leaves as the plant ages. We also included this discussion in the revised manuscript.

Fig S17: (D) Phenotypic progression of uninoculated *B. chinensis* over 25 days, showing signs of chlorosis due to senescence of older leaves as the plant ages.

Lines 371-373: “While some chlorosis was observed in SC@SENDS-treated plants, this was likely due to natural leaf senescence rather than bacterial pathogenicity, as no symptoms of black rot were present (Fig. S17D).”

7. The author chose 5% SDS as a positive control in the biocompatibility test. However, SDS is too harmful for plants, even at low concentrations. Using 5% SDS is excessive, as it causes immediate and noticeable damage to the leaves. It would be better to select a milder positive control. Additionally, conducting tests with various concentrations of SDS, especially at higher levels, would provide more valuable information.

[Authors]: We appreciate the reviewer's feedback on refining our biocompatibility test. We acknowledge that SDS is highly detrimental to plants, even at low concentrations. Our rationale for initially selecting 5% SDS as a positive control was to establish a clear and immediate benchmark for plant stress, allowing us to effectively distinguish the photosynthetic performance of healthy against stressed plants. Such distinction was useful to assess the biocompatibility of SENDS, relative to buffer or SDS treatments. Based on the reviewer's suggestion, we conducted additional tests using 0.5% SDS as a milder positive control. This treatment gradually compromised the photosynthetic performance of plants, as reflected especially in the NPQ profile of treated plants over 7 days. This trend was not observed with SENDS treatment, further highlighting the biocompatibility of SENDS. We have included these results in the revised manuscript. While we recognize the potential value of testing SDS at higher concentrations as the reviewer suggested, we respectfully believe this falls outside the primary scope of our work, as 5% SDS was already sufficient to significantly compromise plants' photosynthetic ability.

We have added the following results as a new **Figure S20** in the revised manuscript:

Fig. S20. Chl-a measurements for plants treated with 0.5% SDS (A) False color maps of sampled leaves from treated plants illustrating the distribution of F_v/F_m , Φ PSII and NPQ values across the surface of the leaves. Evolution of averaged (B) F_v/F_m , (C) Φ PSII and (D) NPQ of sampled leaves across the twenty-day experimental duration. Error bars indicate s.d. (n=3). Statistical differences were calculated using two-sample t-test. ****P<0.0001. **P<0.01, *P<0.05.

8. The manuscript should include a discussion of the feasibility of applying this technology in practice, considering factors such as cost, large-scale production, and environmental impacts.

[Authors]: We appreciate the reviewer’s feedback and have incorporated a discussion on the practical considerations of our technology in the revised manuscript:

Lines 498-511: “In the future, the porous framework of SENDS can be leveraged to encapsulate and deliver other cargoes, including pesticidal compounds, antimicrobial peptides, and double-stranded RNAs (dsRNA), which show great promise for crop protection against pathogens⁷⁷. While SENDS demonstrate significantly enhanced effectiveness over non-targeted nanocarriers, future studies should evaluate their potential to reduce agrochemical use and associated costs while maintaining efficacy. The reliance of SENDS on antibody-based targeting may also pose

cost challenges for large-scale production. Exploring the efficiency of alternative stomata-targeting ligands, such as small nanobodies or non-protein-based moieties, could further improve scalability and affordability. Coupled with advancements in large-scale MOF production, the SENDS platform presented here can guide the design of more cost-effective stomata-targeting nanocarriers for agricultural applications⁷⁸. Additionally, future studies should also evaluate these nanocarriers in real-world field conditions, assessing their efficacy and environmental impact to ensure their effectiveness and safety in large-scale deployment.”

9. How long do SENDS remain attached to the leaf surface and the guard cells of the stomata? The imaging analysis was performed after a gentle rinsing. How strongly does SENDS bind to the guard cells of the stomata? Factors such as rain and wind conditions should be considered in field applications.

[Authors]: We thank the reviewer for the feedback and agree that the localization and potential loss of SENDS to the soil is an important consideration for assessing its suitability under practical situations. As detailed in the response to comment #5 above, we performed rainfastness studies of the nanoparticles and have included this new data in the manuscript as **Figure S10** and the new discussion in **Line 265-274**.

Fig. S10. Rainfastness studies. (A) Schematic detailing rainfastness experimental protocol. Rainfastness of ZIF-8 and SENDS under (B) 2.5 mm and (C) 5 mm rainfall.

Line 265-274: “Beyond stomata localization, we next assessed how rainfall might impact nanoparticle adhesion, which is crucial for practical applications where external environmental factors may affect the performance of nanomaterials on the leaf surface. To study this, we

evaluated the rainfastness of ZIF-8 and SENDS under two simulated rainfall conditions (2.5 mm and 5 mm), using methods previously reported in other literature^{11,50,51} (Fig. S10A). Results showed that under 2.5 mm rainfall, SENDS exhibited approximately 10% higher adhesion (86%) compared to ZIF-8 (75%) (Fig. S10B). A potential explanation for this discrepancy could be the hydrogen bonding interactions between the biomolecules on the surface of SENDS and compounds in the leaf cuticle waxes^{11,52,53}, which may enhance the adhesion of SENDS on the leaf surface. However, at 5 mm rainfall, the adhesion of ZIF-8 and SENDS was similar, at approximately 60% to 70% (Fig. S10C).”

10. SENDS demonstrated 20 times greater effectiveness than a non-targeted nanocarrier in reducing the colonization of *Xanthomonas*. The implications of this finding should be discussed in more detail. How can this advantage of nanotechnology be leveraged in agriculture? Is a 20-fold increase in effectiveness sufficient? A 20-fold increase in effectiveness at a certain concentration does not necessarily imply that a 20-fold lower concentration of the active reagent would yield the same result. It's important to note that achieving this level of targeting requires additional modifications and functionalization, which involve expensive proteins. It is recommended to include a discussion on how the 20-fold higher effectiveness can still be beneficial.

[Authors]: We thank the reviewer for highlighting the practical implications of SENDS' efficacy demonstrated in this work. We acknowledge that while the observed 20-fold reduction in pathogen colonization may not directly translate to a proportional decrease in the required concentration, it provides strong evidence that targeting stomata to reduce pathogen colonization is a viable strategy that can be considered when designing advanced nano-pesticides. We have emphasized this in the concluding statement of our work, (**Lines 516-518**): “*This could pave the way for more precise and effective nanotechnology-based strategies to enhance plant disease resistance, accelerating progress in advancing agricultural productivity and sustainability.*” Additionally, we recognize the need for further studies to clarify the direct implications of SENDS' stomata targeting effects on its potential to reduce agrochemical use and its associated costs, which we have addressed in the updated manuscript:

Lines 501-507: “While SENDS demonstrate significantly enhanced effectiveness over non-targeted nanocarriers, future studies should evaluate their potential to reduce agrochemical use and associated costs while maintaining efficacy. The reliance of SENDS on antibody-based targeting may also pose cost challenges for large-scale production. Exploring the efficiency of alternative stomata-targeting ligands, such as small nanobodies or non-protein-based moieties, could further improve scalability and affordability.”

Minor comments

- Phosphotungstic acid staining was mentioned in the text. Indicate only Figures S1 and S2, but Figure 2 B should also be indicated. It is recommended to give a separate alphabet number to each SEM and TEM image.

[Authors]: We appreciate the reviewer's feedback and have incorporated the suggested changes by calling Figure 2B where phosphotungstic staining was mentioned. However, we respectfully believe that introducing separate alphabets for each TEM and SEM image, particularly in Figure 2, would make it overly cluttered, as it already contains many parts.

- Figure 2C shows that stimulated ZIF-8 is not necessary.

[Authors]: We appreciate the reviewer's feedback. However, we respectfully believe that there is value in including this reference such that readers who are unfamiliar with ZIF-8 would be convinced of its identity and our successful synthesis of this material.

- In lines 139, 148: A histidine-tagged protein might mean polyhistidine-tagged (typically His6) proteins. In the manuscript, it is written as if one histidine can form a coordination bonding with transition metal ions. This should be clarified.

[Authors]: We thank the reviewer for bringing this concern to our attention, and have made this clearer in the revised manuscript:

Lines 137-139: "The stomata-targeting functionality is imparted to ZIF-8 through a modular assembly process which consists of two key steps (Fig. 2A): (i) bioconjugation of polyhistidine-tagged protein G (His6-pG) to ZIF-8..."

Lines 153-155: "This conjugation is facilitated by Lewis acid-base interactions between the imidazole functional groups in the pG polyhistidine tags (Lewis base) and the coordinatively-unsaturated Zn^{2+} sites (Lewis acid) on ZIF-8⁴⁰"

- In line 170, the text mentions the hydrodynamic diameter of SENDS, but no data is shown. Data should be added.

[Authors]: We have added the hydrodynamic diameter data to **Fig. S1** in the revised manuscript:

Fig. S1. (E) Increased hydrodynamic diameter of SENDS following biomolecule functionalization. Error bars indicate s.d. (n=3). Statistical differences were calculated using two-sample t-test. ***P<0.001.

- In line 177, the author mentions the BCA assay in the text at line 177, but the data are not shown or indicated.

[Authors]: We thank the reviewer for bringing this to our attention and have incorporated these results in the revised manuscript. In addition to existing BCA data on cubic SENDS, we also quantified the binding capacity of ZIF-8 in its classical dodecahedral morphology. Results showed that cubic ZIF-8 has a 45% higher IgG binding capacity compared to dodecahedral ZIF-8, thus providing substantiation for our claim that cubic nanoparticles have higher surface area for protein adsorption than dodecahedral geometry (**Lines 128-138**). These results have been incorporated as part of **Figure S1**:

Fig. S1. (F) A comparison of ZIF-8 IgG binding capacity for dodecahedral and cubic morphologies. Error bars indicate s.d. (n=3). Statistical differences were calculated using two-sample t-test. **P<0.01.

- The experiment for the orientation of IgG is not easy to understand. The text indicates Fig. S3, but more explanation is needed in the main text to help readers understand.

[Authors]: We thank the reviewer for this feedback. We have provided a more detailed explanation in the main text and supplemented **Fig. S4** with additional schematics to improve the clarity of this approach.

Lines 189-196: “Fc-specific FITC and Fab-specific Cy3-labeled IgG were added in equal concentrations to ZIF-IgG and SENDS. We hypothesized that the oriented attachment of IgG on SENDS would enhance the accessibility of Fab sites and lead to increased binding of the Fab-specific Cy3-labeled IgG when compared to ZIF-IgG. Upon washing and centrifugation to remove unbound secondary antibodies, the ratio of Cy3 (Fab-specific) to FITC (Fc-specific) fluorescence in SENDS was approximately 64% higher compared to that in ZIF-IgG (Fig. 2F), suggesting a greater availability of the target-recognizing Fab relative to Fc regions.”

Fig. S4. LM6 IgG orientation studies. (A) Schematic detailing experimental procedure to assess IgG orientation on the nanoparticle surface. (B) Schematic describing how Cy3: FITC ratios can be used to assess the accessibility of Fab and Fc regions.

- Enlarging a portion of the confocal micrographs of SENDS in Figure 3C can be confusing and may not be necessary. If this presentation approach is to be used, a description should be included in the figure caption.

[Authors]: We thank the reviewer for this feedback. Our rationale of providing a magnified view of the confocal micrographs is to highlight the stomata and SENDS localization around them, as readers unfamiliar with leaf cellular structures may find it difficult to identify the stomata from the micrographs. To improve the clarity of this presentation, we have edited the figure caption:

Fig. 3. Stomata localization of SENDS on the leaf epidermis. (A) Targeting mechanism of SENDS. (B) Average colocalization percentage of SENDS on *A. thaliana*, *V. faba* and *B. chinensis* based on CLSM micrographs. Error bars indicate s.d. (n = 5). Statistical differences were calculated using two-sample t-test. ****P<0.0001. (C) CLSM (scale bar = 20 μ m) and SEM micrographs (scale bar = 10 μ m) showing the localization of ZIF-8 and SENDS on the leaf epidermis of *A. thaliana*. CLSM inset: magnified view of stomata and SENDS localization pattern.

Reviewer #2 (Remarks to the Author):

Summary

The manuscript presents an antimicrobial loaded MOF, decorated with functional antibody LM6 to target the stomata of crops and prevent bacterial infection. This work leverages advances in the biomedical field, material science, and plant bionanotechnology to create a sound and working material that populates the stomata after foliar spray and reduces the disease index of bacterial infection. Such work is considered to be a significant improvement over conventional agrochemicals and is a subject of interest to experts in many fields of study. The manuscript is well written, using proper English, and contains compelling figures, which will please the readers of the journal.

[Authors]: We highly appreciate the reviewer's positive comments on the significance and presentation quality of our work.

Although this work presents a good advancement in the field, several improvements need to be addressed before the publication of this manuscript.

Introduction

1. Although the introduction is well written and contains relevant references, there is no mention of the extensive use of antibodies, such as LM6, for immunohistological studies of plant cell walls. These have been utilized for over 15 years to target the delivery of gold nanoparticles to visualize specific tissues under electron scanning microscopy. The evidence of these biorecognition molecules for such studies is robust and not exclusive to the work of nano-enabled agriculture. Such work should be included and properly cited somewhere in lines 59-68.

[Authors]: We thank the reviewer for this suggestion and have revised the introduction to mention antibody-targeting approaches in plant studies:

Lines 64-73: "Beyond targeting subcellular organelles, strategies for directing nanoparticles to specialized plant cellular structures presents new opportunities in plant engineering. Targeted delivery using antibodies that recognize specific carbohydrates in the plant cell wall has been instrumental in studying cell wall structure and functionality^{23,24}. This approach has been widely validated, as demonstrated by immunogold labeling in electron microscopy to visualize plant cell wall structure and composition^{25,26}. Leveraging similar biorecognition strategies could enable precise nanoparticle delivery to important epidermal leaf structures, such as stomata, that are accessible through foliar application. This delivery method complements organelle-targeting strategies typically achieved through manual mechanical infiltration, thereby facilitating scalable precision delivery systems in agriculture."

Results

2. Some results are written in the present tense instead of the past tense. E.g. Line 272. "...while the MIC of SC encapsulated in 25 $\mu\text{g}/\text{mL}$ ZIF-8 is 7 $\mu\text{g}/\text{mL}$ "

[Authors]: We thank the reviewer for bringing this to our attention and have made the necessary corrections in the revised manuscript.

3. By definition, the minimum inhibitory concentration (MIC) is the minimum concentration of an active required to inhibit the growth of a bacteria, based on the experiment this should be the concentration that prevents the bacteria from growing a CFU/mL higher than the initial inoculum. I believe that what is being described in the results is the minimum bactericidal concentration (MBC), which is the lowest concentration necessary to kill the bacteria, as evidenced by the CFU assay.

[Authors]: We appreciate the reviewer's feedback to clarify the distinction between MIC and MBC. We have edited the manuscript accordingly to make it clearer to the readers:

Lines 297-300: "To determine the optimal composition of SENDS for effectively inhibiting the proliferation and bacterial activity of *X. campestris*, colony forming unit (CFU) assays were conducted to assess the minimum bactericidal concentration (MBC) of pure ZIF-8, SC and SC@ZIF."

4. Line 330: "While some chlorosis was observed in SC@SENDS-treated plants, this was likely due to natural leaf senescence rather than bacterial pathogenicity, as no symptoms of black rot were present." This is observed in Fig 5 E. Although it might be reasonable to assume this, the chlorosis is not visible in Fig 6. (biocompatibility) or recorded to affect the photosynthetics of the leaves. Therefore, this chlorosis is most likely due to bacterial inoculation. To conclusively determine the cause of the chlorosis, additional observation time (dpi 25 or 30), as well as an untreated and non-inoculated control are needed.

[Authors]: We thank the reviewer for this suggestion. We have conducted additional experiments and recorded phenotypic progression of uninoculated plants over 25 days, which showed similar leaf yellowing as SC@SENDS-treated plants under the growth conditions we employed. Specifically, uninoculated plants also showed signs of chlorosis from senescence of older leaves as the plant ages. We have included this additional data as **Fig. S17D** in the revised manuscript.

Fig S17: (D) Phenotypic progression of uninoculated *B. chinensis* over 25 days, showing signs of chlorosis due to senescence of older leaves as the plant ages.

Lines 371-373: “While some chlorosis was observed in SC@SENDS-treated plants, this was likely due to natural leaf senescence rather than bacterial pathogenicity, as no symptoms of black rot were present (Fig. S17D).”

5. Line 316: “This underscores the critical role of SENDS’s stomata-targeting capability in enhancing plant resistance to pathogen invasion without disrupting normal stomatal function.” Although SC@SENDS significantly improved the disease index, the effect of sanguinarine on stomatal opening was not assessed. Therefore, it is possible that SC@SENDS is causing stomatal closure and consequently, reducing bacterial colonization in the stomata. A SC@SENDS and sanguinarine control need to be included in the stomatal aperture experiment.

[Authors]: We thank the reviewer for this suggestion and agree that incorporating these additional controls would further strengthen our claim regarding the antibacterial mechanism of SENDS. We performed additional experiments to assess the effect of SC and SC@SENDS on stomata aperture and found that neither treatment induced stomatal closure, as demonstrated by the stomatal aperture data now included in **Fig. S15** shown below. Discussion on this additional data has also been incorporated in **Line 330-334** in the revised manuscript.

Fig. S15 Stomatal aperture measurements. Micrographs depicting stomata of leaves treated with (A) buffer and (B) ABA, resulting in larger and smaller average stomatal apertures respectively. Scale bars represent 20 μm . (C) Stomatal aperture measurements for SC and SC@SENDS treated leaves. Error bars indicate s.d. (n=50). Statistical differences were calculated using ANOVA with Tukey’s post-hoc test. No significant difference was observed between the tested groups, confirming that SC and SC@SENDS did not induce stomatal closure.

Line 330-334: “The average stomatal aperture in SENDS-treated leaves was similar to that in leaves treated with MES buffer, indicating that SENDS did not induce stomatal closure (Fig. 5C). A similar trend was observed in leaves treated with SC and SC@SENDs (Fig. S15C), confirming that the observed reduction in pathogen internalization is due to the antibacterial properties and stomata-targeting capability of SC@SENDs.”

6. Additionally, proper controls should be included in the in vivo antimicrobial efficacy assays. These include an untreated and uninoculated control, as well as a sanguinarine treated plant group. Moreover, a commercial product based on plant alkaloids such as a sanguinarine-based pesticide should also be used as a comparison to contextualize the advancement of the SENDS technology.

[Authors]: We fully agree with the reviewer and have conducted additional in vivo experiments to quantify the bacterial load in the apoplast of plants treated with unencapsulated SC at the same concentration as in SC@SENDs. Our results showed that free SC did not significantly reduce bacterial load, whereas SC@SENDs achieved a 2.5 to 3 order of magnitude greater reduction compared to free SC, further highlighting the benefits of encapsulation and targeted delivery provided by SENDS. We also incorporated uninoculated controls, which showed no bacteria growth, confirming that plants were not infected prior to the experiment. These new results have been added as an extension to **Fig. S13**.

Furthermore, to the best of our knowledge, there is currently no commercial alkaloid-based product for use in our country. Many commercial products also combine alkaloids with other active ingredients or adjuvants, making comparison with SENDS unfair as these adjuvants and ingredients may exhibit additional antimicrobial activity. Hence, we believe that comparison with free SC in this new data (**Fig. S13E**) is the most appropriate to contextualize the significance of our SENDS technology.

Fig. S13. (C) Uninoculated drop plate assay control showing no CFU growth. (D) Drop plate assay for free-SC treated *B. chinensis* with 10^8 CFU/mL *X. campestris*. (E) Average bacterial load in leaf apoplast of free SC-treated plants in comparison to untreated and SC@SENDs treated plants. Error bars indicate s.d. (n=5). Statistical differences were calculated using two-sample t-test. *** $P < 0.001$, **** $P < 0.0001$.

Lines 322-325: “Stomata-targeting SC@SENDS reduced the internalization of *X. campestris* (bacteria counts per mm² of leaf) by approximately 400-fold, 200-fold and 20-fold compared to untreated, SC and SC@ZIF-treated plants, respectively (Fig. 5B, S13, S14).”

7. Although SC@sanguirine reduced the disease index no micrographic evidence was recorded of the material interaction directly with the bacterial. SEM or confocal microscopy should be employed to visualize how SENDS is preventing bacteria from colonizing the stomata. Appropriate controls should be assessed too, like untreated controls to visualize infection.

[Authors]: We appreciate the reviewer’s suggestions and strongly agree that visualizing bacterial invasion through the stomata and the antibacterial effect of SENDS would further strengthen our claims. We performed imaging experiments to visualize live and dead bacteria (*Xcc*) under a confocal microscope. In the untreated control, bacteria were observed congregating at the stomatal pore, with most appearing viable (green), confirming that *Xcc* could colonize plants through the stomata. In contrast, in samples treated with SC@SENDS, bacteria in the vicinity of the stomata had been effectively killed (red), further highlighting the antibacterial efficacy of SENDS around the stomata. Additionally, quantification of the live and dead bacteria in the vicinity of stomatal pores revealed a reduction in bacteria viability of up to 70% due to SENDS treatment. We have incorporated these results into **Figure 5D** and **Figure S16** in the revised manuscript.

Fig. 5. (D) Confocal micrographs illustrating *X. campestris* entry to the stomata and the antimicrobial activity of SC@SENDS towards *X. campestris* surrounding the stomata. Scale bars represent 10 μ m.

Fig. S16. Visualizing the antibacterial activity of SC@SENDS. (A) Additional confocal micrographs illustrating *X. campestris* entry to the stomata and the antimicrobial activity of SC@SENDS towards *X. campestris* surrounding the stomata. Scale bars represent 10 μm . (B) Analysis of bacterial viability surrounding stomata based on confocal micrographs. Error bars indicate s.d. (n=5). Statistical differences were calculated using two-sample t-test.

Lines 354-360: “Confocal microscopy, combined with the Live/Dead (SYTO 9/propidium iodide) assay was used to visualize the antimicrobial activity of SC@SENDS around the stomata. Untreated samples showed colonization of *X. campestris* around the stomata, with most bacteria appearing green, indicating viability. In contrast, SC@SENDS treated samples demonstrated a significant reduction in bacterial viability in the vicinity of the stomata, with dead bacteria appearing red (Fig. 5D). Quantification of live and dead bacteria by manual counting revealed up to a 70% reduction in bacterial viability around the stomata following SC@SENDS treatment. (Fig. S16B)”

Discussion

8. Line 443: “By precisely targeting the stomatal guard cells, the primary entry points for pathogens into the apoplast, we demonstrated a nanotechnology-based approach to confer plants with enhanced protection against pathogen internalization...” This is not factual. For example, infections by *Xanthomonas campestris* (the pathogen used) on leaves have been shown to start at the hydathodes, at the edge of the leaf. Please see the following: <https://doi.org/10.1016/j.cub.2023.01.013> and <https://doi.org/10.1104/pp.16.01852> Furthermore, this is backed by symptoms used to establish the disease index in this work, as DI 1 and 2 mainly present chlorosis at the edges of the leaf. Considering this, it may well be that SC@SENDS protects the hydathodes rather than the stomata. LM6 antibody does not target specifically guard cells, but moieties in pectin that are present in several leaf tissues, including hydathodes.

[Authors]: We appreciate the reviewer for this feedback. We acknowledge that while the preferred route of entry for *Xcc* may indeed be the hydathodes, our results, along with other studies, suggest that *Xcc* can also enter the plant through the stomata. Using confocal microscopy, we observed the congregation of this bacteria around the stomata of infected plants (**Fig. 5D and Fig. S16** in the revised manuscript), supporting the notion that stomatal entry is also a viable infection route. Taken together, our findings suggest that targeting the stomata can effectively reduce bacterial pathogenicity. Moreover, previous studies have demonstrated that *Xcc* can bypass natural stomatal defenses by actively reverting stomatal closure using a virulence factor, facilitating their entry^{6,7}. We acknowledge that, in addition to stomata, some nanoparticles may also localize to other leaf tissues such as hydathodes, which warrant further investigation in future studies. Nonetheless, our current data shows that SENDS can localize on the stomatal guard cells and overall support the conclusion that targeting stomata provides effective protection.

We appreciate the reviewer’s constructive feedback and have updated the manuscript to more accurately describe pathogens’ entry routes, while maintaining that targeting the stomata is an effective strategy for enhancing plant protection:

Lines 485-489: “While *X. campestris* may also enter through the hydathodes to initiate virulence⁷⁶, our results demonstrate that stomata-targeting nanoparticles could provide an effective strategy to enhance plant resistance to bacterial invasion. By precisely targeting the stomatal guard cells, this nanotechnology-based approach offers enhanced protection against pathogen internalization, without altering the natural stomatal movement in response to environmental cues.”

9. Line 456: “Overall, this study presents a breakthrough in plant nanotechnology by introducing a rationally designed nanocarrier platform that precisely targets stomata to bolster plant defenses against pathogen colonization.” This approach has been reported before (as referenced in the manuscript) and therefore does not qualify as a breakthrough. Additionally, gold nanoparticles tagged with these antibodies have been significantly reported for immunolabeling studies and visualization of rhamnogalacturonans for over 15 years. A more modest description of the work is suggested, as the study is a methodological improvement of previous studies using MOFs.

[Authors]: We thank the reviewer and appreciate the opportunity to clarify our scientific contribution. While targeted delivery using antibodies is a well-established technique, our study builds on this approach by showing how targeting specific plant cells or tissues can improve plant defense. Specifically, we showed that targeting the stomata with a MOF-based nanocarrier system can enhance plant resistance against pathogen colonization. We have revised the manuscript to clarify this:

Lines 636-638: “Overall, this study advances plant nanotechnology by developing a rationally designed MOF-based nanocarrier platform for stomata-targeted delivery, demonstrating its potential to bolster plant defense against pathogen colonization.”

We have also revised the introduction to discuss antibody-targeting approaches in plant science:

Lines 64-73: “Beyond targeting subcellular organelles, strategies for directing nanoparticles to specialized plant cellular structures presents new opportunities in plant engineering. Targeted delivery using antibodies that recognize specific carbohydrates in the plant cell wall has been instrumental in studying cell wall structure and functionality^{23,24}. This approach has been widely validated, as demonstrated by immunogold labeling in electron microscopy to visualize plant cell wall structure and composition^{25,26}. Leveraging similar biorecognition strategies could enable precise nanoparticle delivery to important epidermal leaf structures, such as stomata, that are accessible through foliar application. This approach complements organelle-targeting strategies typically achieved through manual mechanical infiltration, thereby facilitating scalable precision delivery systems in agriculture.”

10. Although outside the scope of the study, there is no mention of the scalability or feasibility of using antibodies for agricultural purposes. It is appropriate to include one or two lines for non-experts to understand that such refined architecture and biotargeting motifs are not yet practical for mass production.

[Authors]: We fully agree with the reviewers on this and have included the following discussion in our manuscript:

Lines 498-511: “In the future, the porous framework of SENDS can be leveraged to encapsulate and deliver other cargoes, including pesticidal compounds, antimicrobial peptides, and double-stranded RNAs (dsRNA), which show great promise for crop protection against pathogens⁷⁷. While SENDS demonstrate significantly enhanced effectiveness over non-targeted nanocarriers, future studies should evaluate their potential to reduce agrochemical use and associated costs while maintaining efficacy. The reliance of SENDS on antibody-based targeting may also pose cost challenges for large-scale production. Exploring the efficiency of alternative stomata-targeting ligands, such as small nanobodies or non-protein-based moieties, could further improve scalability and affordability. Coupled with advancements in large-scale MOF production, the SENDS platform presented here can guide the design of more cost-effective stomata-targeting nanocarriers for agricultural applications⁷⁸. Additionally, future studies should also evaluate these nanocarriers in real-world field conditions, assessing their efficacy and environmental impact to ensure their effectiveness and safety in large-scale deployment.”

Materials and Methods

11. Several details regarding the nanoparticle localization and imaging experiments need to be specified. This includes: The equation utilized to quantify the material’s colocalization of the stomata, the area (mm²) assessed on the leaves, the volume/concentration of SEND and Zif utilized.

[Authors]: We thank the reviewer for bringing this to our attention and have included the requested information in the revised manuscript:

Line 594: “F@ZIF-8 and F@SENDS (2 mg/mL) suspended in 10 mM, pH 7.5 MES buffer (5 mL) were sprayed onto the abaxial side of the leaves of all three plant species.”

Line 600-603: “An intensity threshold was applied to the GFP channel and colocalized fluorescent signals (fluorescein) were quantified as a percentage of total fluorescent signals in a single field of view (Equation 1). The total area assessed for each leaf sample was approximately 0.225 mm²”

$$\text{Colocalization (\%)} = \frac{\text{Total colocalized fluorescence intensity}}{\text{Total fluorescence intensity}} \times 100\% \quad (1)$$

12. Additionally, it is unclear why the treatments are rinsed before assessing the colonization. This leads to the removal of the loosely adhered material, leaving only the particles that are adhered to the stomata due to the antibody. In turn, this can skew the quantification of the colonization of material.

[Authors]: We appreciate the reviewer’s feedback and would like to clarify that the rinsing step was implemented to remove loosely adhered material that non-specifically contributes to background signal and obscure stomata-targeting effects. Rinsing ensures that the observed localization primarily reflects particles that interact with the leaf surface rather than those that may have settled non-specifically without meaningful adhesion. This distinction is important for

accurately assessing stomata-targeting interactions versus nonspecific accumulation of the nanoparticles. Furthermore, we performed additional rainfastness studies that demonstrated similar adhesion between ZIF-8 and SENDS under high rainfall conditions (**Fig. S10** in the revised manuscript), suggesting that rinsing is unlikely to affect our conclusion. Additionally, rinsing provides a more realistic measure of SENDS' stomata-targeting effect under conditions where weakly-adhered particles would likely be lost to environmental factors such as rain or irrigation. Given these considerations, we believe that rinsing is the most appropriate approach to ensure meaningful interpretation of the data, particularly in assessing the true adhesion and localization of SENDS at the stomata.

The data and discussion of the rainfastness study has been included in the revised manuscript:

Lines 265-274: “Beyond stomata localization, we next assessed how rainfall might impact nanoparticle adhesion, which is crucial for practical applications where external environmental factors may affect the performance of nanomaterials on the leaf surface. To study this, we evaluated the rainfastness of ZIF-8 and SENDS under two simulated rainfall conditions (2.5 mm and 5 mm), using methods previously reported in other literature^{11,50,51} (Fig. S10A). Results showed that under 2.5 mm rainfall, SENDS exhibited approximately 10% higher adhesion (86%) compared to ZIF-8 (75%) (Fig. S10B). A potential explanation for this discrepancy could be the hydrogen bonding interactions between the biomolecules on the surface of SENDS and compounds in the leaf cuticle waxes^{11,52,53}, which may enhance the adhesion of SENDS on the leaf surface. However, at 5 mm rainfall, the adhesion of ZIF-8 and SENDS was similar, at approximately 60% to 70% (Fig. S10C).”

Fig. S10. Rainfastness studies. (A) Schematic detailing experimental protocol. Rainfastness of ZIF-8 and SENDS under (B) 2.5 mm and (C) 5 mm rainfall.

13. Several details are noticed in the antibacterial in vitro studies that can significantly affect the results. The recommended media for the bacteria is YGC medium, which was not utilized. This can lead to a reduced count of colonies forming on the agar. Furthermore, the initial inoculum of bacteria is relatively low at 10^4 CFU/mL. It is recommended to perform the antimicrobial assessment at a bacterial load of 10^5 - 10^6 CFU/mL. This can also lead to a lower colony count. Finally, the antimicrobial treatments and the bacteria are suspended in DI water, which in itself is not conducive to bacterial growth. This can also lead to a lower colony count. It is recommended to perform this assay using YGC broth.

[Authors]: We appreciate the reviewer's feedback and understand the concerns regarding medium selection and bacterial inoculum concentration in our in-vitro antibacterial studies. Although YGC is the recommended medium for *Xanthomonas campestris pv. campestris* (Xcc), we opted to culture Xcc in DI water and on ISP agar after encountering challenges with YGC during initial trials. Specifically, the recommended YGC formulation for the Xcc strain we are currently using (ATCC medium 73) includes insoluble calcium carbonate, which led to uneven bacterial spreading and made CFU enumeration challenging, potentially introducing variability and inconsistencies in our results. Although ISP agar may not be the optimal medium for maximizing bacterial growth, it provided a more consistent platform for assessing the antibacterial efficacy of the different treatments.

To address concerns regarding bacterial load and the use of YGC agar, we conducted additional experiments using 10^6 CFU/mL inoculation on YGC agar. These experiments demonstrated that the minimum bactericidal concentration (MBC) of 7 μ g/mL SC in 25 μ g/mL ZIF-8 remains effective. This new result has been incorporated as part of **Figure S14**.

Fig. S14. (A) CFU assay for *X. campestris* conducted at 10^6 CFU / mL initial inoculum on YGC agar. No CFU growth was observed when incubated with 7 μ g/mL SC in 25 μ g/mL ZIF-8, indicating that MBC is maintained despite a higher initial inoculum and changes in growth media.

14. For the pathogen infection and bacterial load quantification experiments, the concentration of SC@SENDS is not described and the volume of both SENDS and bacteria is not specified.

[Authors]: We thank the reviewers for bringing this to our attention and have included this information in the revised manuscript:

Line 645-648: “SC@ZIF and SC@SENDS (3 mg/mL) suspended in 10 mM, pH 7.5 MES buffer (5 mL) were sprayed onto the abaxial side of *B. chinensis* leaves and left undisturbed for 24 hours. The plants were then infected by spraying the abaxial side with 10^8 CFU/mL *X. campestris* in a 5 mL 0.05% Silwet-77 solution.”

15. Furthermore, ethanol was used to rinse the leaves which can penetrate apoplast and reduce bacteria CFU. The use of wipes soaked in waterborne antiseptics, such as bleach, is recommended given that water is too viscous to readily penetrate the apoplast.

[Authors]: We appreciate the reviewer’s suggestion and understand the concern regarding the potential impact of ethanol on quantified bacterial load by penetrating the apoplast. We would like to clarify that instead of rinsing the leaves in ethanol, we gently wiped the leaf surface with wipes soaked in 70% ethanol. This has been corrected in the revised manuscript:

Lines 651-654: “For quantification of bacterial load in the apoplast of the plants, mature, fully expanded leaves from treated plants of the same growth stage were selected by numbering leaves and chosen using a random number generator. The leaves were gently wiped with ethanol-soaked wipes to remove surface contamination.”

To address the concern that ethanol may lead to artefacts in the quantification of apoplastic bacterial population, we conducted additional experiments comparing the effects of ethanol and bleach-soaked wipes on bacterial CFU counts. Our results indicate that there is no significant difference between the two methods, as shown by the following data. We have included this data in the revised manuscript as **Figure S14B**.

Fig. S14: (B) Drop plate assay quantifying the apoplastic bacterial load following surface sterilization using two different compounds, ethanol and bleach. No significant difference in quantified bacterial load was observed.

16. There are no criteria described for sampling the leaf tissue. This can lead to skewed results as tissue in the edge of the leaves can have higher inoculum compared to tissue in the center.

Considering this, the number of samples for the bacteria apoplast bacteria quantification is rather small with $n=3$. There is no description of whether different leaves were sampled or a singular leaf at different areas. If few leaves are sampled, this can also skew the results, as healthy/sick leaves can be preferentially sampled. The experiments should also include an untreated and uninoculated control, as well as a sanguinarine treated plant group.

[Authors]: We thank the reviewer for bringing this to our attention and have edited the protocol to reflect the sampling techniques:

Lines 651-658: “For quantification of bacterial load in the apoplast of the plants, mature, fully expanded leaves from treated plants of the same growth stage were selected by numbering leaves and chosen using a random number generator. The leaves were gently wiped with ethanol-soaked wipes to remove surface contamination. Using a 4 mm circular cutter, 4 leaf discs were excised approximately 1 cm from the midrib, avoiding large veins and the leaf edges. Two leaf discs from each side of the midrib were sampled, yielding a total of 4 leaf discs, each with an area of 12.6 mm^2 . The leaf discs were placed into a 2 mL Eppendorf tube containing $200 \mu\text{L}$ of sterile DI water and thoroughly crushed using the back end of a plate spreader, constituting one replicate.”

We also increased the number of data points presented from 3 to 5. This is reflected in changes to **Fig. 5B**, as well as the new experiments involving the SC only treated group. Assays from the uninoculated control samples showed no CFU growth. The untreated control has previously been included in the Fig. 5B.

Fig. S13. (C) Uninoculated drop plate assay control showing no CFU growth. (D) Drop plate assay for free-SC treated *B. chinensis* with 10^8 CFU/mL *X. campestris*. (E) Average bacterial load in leaf apoplast of free SC-treated plants in comparison to untreated and SC@SENDS treated plants. Error bars indicate s.d. ($n=5$). Statistical differences were calculated using two-sample t-test. *** $P<0.001$, **** $P<0.0001$.

17. For phenotyping tests, no uninoculated control was included. Similarly, 3 leaves are not enough to evaluate an entire plant, unless specific criteria are set for sampling (such as sampling leaves from the same plant growth stage).

[Authors]: We thank the reviewer for this suggestion and have added the phenotypic progression of an uninoculated plant over 25 days to **Fig. S17**:

Fig S17D: (D) Phenotypic progression of uninoculated *B. chinensis* over 25 days, showing signs of chlorosis due to senescence of older leaves as the plant ages.

Line 665-672: “For phenotyping tests, plants were treated and left under a 14-hour light, 10-hour dark photoperiod with 50% relative humidity, and a constant temperature of 23°C in the plant chamber. Photographs were taken at 10 and 20 dpi. The overall plant disease index for a single plant was calculated by averaging the DI scores for all leaves. Replicates were taken from plants of the same growth stage. Chl-a measurements were conducted using the Fluorcam 1300 (Photon Systems Instruments) during the same light / dark photoperiod each day to ensure consistency. Mature, fully expanded leaves from treated plants of the same growth stage were selected by numbering leaves and chosen using a random number generator.”

Reviewer #3 (Remarks to the Author):

General comments- This paper presents a rationally designed nanocarrier platform named surface ligand-engineered nanoparticles for targeted delivery to stomata (SENDS) for targeted delivery of antibiotics to stomata of a few common crops to enhance plant defense against pathogens. A stomata targeting antibody IgG was attached to the surface of ZIF-8 MOF core through modular assembly. The antibody functionalized nanocarriers have exhibited 15 times higher colocalization with stomata compared to unmodified MOF. The SENDS was loaded with a common antibiotic sanguinarine chloride (SC) and the SC loaded SENDS have exhibited antimicrobial activity both in vitro and in vivo, with 20 folds higher antimicrobial activity compared to the untargeted ZIF8, highlighting the improved efficacy in plant pathogen control by this targeted delivery strategy. Deployment of this material to stomata did not disturb stomata movements and plant photosynthetic activity. Although the concept of stomata targeting nanoparticles was previously reported, this is the first demonstration of improved antimicrobial efficacy achieved by targeted antibiotic delivery to stomata. The paper is very well written, topic of this paper is timely, and the proposed stomata targeting technology can be a promising and viable solution for plant pathogen control, provided that their scalability and non-toxicity are proven. To increase the impact of the study and to enhance the scientificity of the proposed work, I would recommend addressing the following:

[Authors]: We appreciate the reviewer's positive feedback on the significance and novelty of our work, and for highlighting its potential as a new promising strategy for plant pathogen control.

1. Agriculture in general has a very thin profit margin, therefore agrochemicals need to be very affordable. The MOF and stomata targeting antibody can be very expensive to manufacture and the strategy may be not viable economically. The authors should discuss potential challenges and opportunities for implementing this technology at an agriculture-relevant scale and how to perform future studies that may support the feasibility of the proposed delivery platform.

[Authors]: We appreciate the reviewer's feedback and have incorporated a discussion on the practical considerations of our technology in the revised manuscript:

Lines 498-511: "In the future, the porous framework of SENDS can be leveraged to encapsulate and deliver other cargoes, including pesticidal compounds, antimicrobial peptides, and double-stranded RNAs (dsRNA), which show great promise for crop protection against pathogens⁷⁷. While SENDS demonstrate significantly enhanced effectiveness over non-targeted nanocarriers, future studies should evaluate their potential to reduce agrochemical use and associated costs while maintaining efficacy. The reliance of SENDS on antibody-based targeting may also pose cost challenges for large-scale production. Exploring the efficiency of alternative stomata-targeting ligands, such as small nanobodies or non-protein-based moieties, could further improve scalability and affordability. Coupled with advancements in large-scale MOF production, the SENDS platform presented here can guide the design of more cost-effective stomata-targeting nanocarriers for agricultural applications⁷⁸. Additionally, future studies should also evaluate these nanocarriers in real-world field conditions, assessing their efficacy and environmental impact to ensure their effectiveness and safety in large-scale deployment."

2. The stomata targeting capacity of this material is demonstrated only on three types of dicot plants. Monocot plants, such as maize, wheat and rice also need to be studied to examine the universality and versatility of this technology.

[Authors]: We thank the reviewer for this suggestion and fully agree that demonstrating SENDS' targeting capabilities on monocots could further improve the impact of the work. We performed additional experiments to test SENDS' stomata targeting on two monocot species: *Oryza sativa* (rice) and *Hordeum vulgare* (barley). Confocal microscopy was used to visualize and quantify SENDS localization. Our results show that the colocalization percentage of SENDS in *O. sativa* and *H. vulgare* are 6.5 and 12.4 times higher than ZIF-8 colocalization respectively, thus demonstrating the versatility of SENDS' stomata targeting capability. The confocal micrographs and data for colocalization percentage have been added as **Fig. S8 and S9**, and the main text has been edited to reflect these results:

Lines 98-101: “Confocal and scanning electron microscopy confirm the stomata-targeting capability of SENDS in five different plant species: *Arabidopsis thaliana*, *Brassica rapa* (subsp. *chinensis*) *Vicia faba*, *Oryza sativa* and *Hordeum vulgare*.”

Lines 254-264: “To demonstrate the applicability of our stomata-targeting SENDS across different plant species, we also studied the localization of SENDS on the leaf surfaces of two other dicotyledonous species – *B. chinensis* (pak choy), an economically important vegetable crop in Asia, and *V. faba* (fava bean), a model system in stomatal physiology research⁴⁹ – as well as two monocotyledonous species, *O. sativa* (rice) and *H. vulgare* (barley), both of which are important food crops. Micrographs obtained for all species exhibited similar localization patterns of SENDS on the guard cells of the stomata to that of *A. thaliana* (Figs. 3D, E, S6, S7). Additionally, the higher colocalization percentage of SENDS compared to unfunctionalized ZIF-8 for all four plant species confirmed the stomata-targeting efficacy of SENDS across diverse plant species (Fig. 3B, S9), indicating that targeting arabinan of guard cells can be a conserved strategy for the design of stomata-targeting nanocarriers for different plant species.”

Line 477-480: “This targeting efficiency was consistent across multiple dicotyledonous and monocotyledonous species, including *A. thaliana*, *B. chinensis*, *V. faba*, *O. sativa* and *H. vulgare*, highlighting the broad applicability of this nanocarrier approach.”

Fig. S8. SENDS localization in monocot species. Confocal micrographs exhibiting localization of SENDS for (A) *O. sativa* and (B) *H. vulgare*. Scale bars represent 20 μm .

Fig. S9 Average colocalization percentage of SENDS across all tested plant species on based on CLSM micrographs. Error bars indicate s.d. (n = 5). Statistical differences were calculated using two-sample t-test. ****P<0.0001, **P<0.01

3. Only the untreated plants are used as control for the in-vivo antimicrobial activity study. Another control with free SC at the same concentration is needed to examine the efficacy of this technology compared to conventional application methods.

[Authors]: We fully agree with the reviewer and have conducted additional in vivo experiments to quantify the bacterial load in the apoplast of plants treated with unencapsulated SC at the same concentration as in SC@SENDS. Our results showed that free SC did not significantly reduce bacterial load, whereas SC@SENDS achieved a 2.5 to 3 order of magnitude greater reduction compared to free SC, further highlighting the benefits of encapsulation and targeted delivery provided by SENDS. We also incorporated uninoculated controls, which showed no bacteria growth, confirming that plants were not infected prior to the experiment. These new results have been added as an extension to **Fig. S13**.

Fig. S13. (C) Uninoculated drop plate assay control showing no CFU growth. **(D)** Drop plate assay for free-SC treated *B. chinensis* with 10^8 CFU/mL *X. campestris*. **(E)** Average bacterial load in leaf apoplast of free SC-treated plants in comparison to untreated and SC@SENDS treated plants. Error bars indicate s.d. (n=5). Statistical differences were calculated using two-sample t-test. ***P<0.001, ****P<0.0001.

Line 322-325: “Stomata-targeting SC@SENDS reduced the internalization of *X. campestris* (bacteria counts per mm² of leaf) by approximately 400-fold, 200-fold and 20-fold compared to untreated, SC and SC@ZIF-treated plants, respectively (Fig. 5B, S13, S14).”

4. A rain fastness study for SENDS ZIF-8 is necessary to examine the persistence and efficacy of this technology under more relevant condition.

[Authors]: We thank the reviewer for the feedback and agree that the potential loss of SENDS to the soil is an important consideration for assessing its suitability under practical situations. To assess the adhesion of SENDS under precipitation, we conducted rainfastness tests¹⁻³ on ZIF-8 and SENDS under two simulated rainfall intensities: 2.5 mm and 5 mm. Our results indicate that under 2.5 mm rainfall, SENDS demonstrated roughly 10% higher adhesion compared to ZIF-8 (86% vs 75%). This discrepancy could potentially be due to hydrogen bonding interactions between the biomolecules on the SENDS' surface and compounds in the cuticular waxes of the leaf epidermis^{1,4,5}. At 5 mm rainfall, however, ZIF-8 and SENDS exhibited similar adhesion, suggesting higher rainfall intensities may overcome the initial adhesion advantage of SENDS.

Lines 265-274: “Beyond stomata localization, we next assessed how rainfall might impact nanoparticle adhesion, which is crucial for practical applications where external environmental factors may affect the performance of nanomaterials on the leaf surface. To study this, we evaluated the rainfastness of ZIF-8 and SENDS under two simulated rainfall conditions (2.5 mm and 5 mm), using methods previously reported in other literature^{11,50,51} (Fig. S10A). Results showed that under 2.5 mm rainfall, SENDS exhibited approximately 10% higher adhesion (86%) compared to ZIF-8 (75%) (Fig. S10B). A potential explanation for this discrepancy could be the hydrogen bonding interactions between the biomolecules on the surface of SENDS and compounds in the leaf cuticle waxes^{11,52,53}, which may enhance the adhesion of SENDS on the leaf surface. However, at 5 mm rainfall, the adhesion of ZIF-8 and SENDS was similar, at approximately 60% to 70% (Fig. S10C).”

Fig. S10. Rain fastness studies. (A) Schematic detailing experimental protocol. Rain fastness of ZIF-8 and SENDS under (B) 2.5 mm and (C) 5 mm rainfall.

5. The authors have used CTAB to adjust the shape of ZIF-8 and claimed the change in ZIF-8 morphology from dodecahedral to a cubic shape can improve the oriented assembly of the stomata-targeting biomolecules, hence stomata targeting efficacy of the material. However, CTAB is toxic and can pose risks to ecological and human health. A more compelling justification could be made by comparing the performance of the dodecahedral and cubic-shaped materials, which would strengthen the argument and highlight the advantages of the chosen geometry.

[Authors]: We appreciate the reviewer’s suggestion and agree that stronger justification is needed for the morphological modification of ZIF-8 using CTAB. Our rationale for synthesizing cubic ZIF-8 is to enhance the surface area-to-volume ratio compared to conventional dodecahedral ZIF-8, thereby enabling a higher density of stomata-targeting IgG decoration on the nanoparticle surface, ultimately improving SENDS’ targeting efficiency. To substantiate this claim, we conducted additional experiments to quantify and compare the IgG binding capacity of dodecahedral and cubic ZIF-8 using BCA assay. Our results showed that cubic ZIF-8 exhibits a 45% higher IgG binding capacity than dodecahedral ZIF-8, demonstrating that the cubic morphology enhances functionalization. We have included the following data as part of **Fig. S1**:

Fig. S1. (F) A comparison of ZIF-8 IgG binding capacity for dodecahedral and cubic morphologies. Error bars indicate s.d. (n=3). Statistical differences were calculated using two-sample t-test. **P<0.01.

Line 130-133: “This modification was intended to increase the outer surface area-to-volume ratio of the particles, thereby allowing for the attachment of a higher number of stomata-targeting biomolecules.”

6. Fig. 1 lacks any data and is just a schematic. Perhaps it should be removed or integrated with Fig. 2.

[Authors]: We thank the reviewer for the feedback. Fig. 1 serves as an introductory schematic to provide context for the reader and enhance the understanding of the following figures. As it plays an important role in the overall clarity of the manuscript, we have decided to retain it rather than integrating it with Fig. 2. We believe that this can offer a clearer understanding of our overall approach to the readers.

Reviewer #4 (Remarks to the Author):

[Authors]: We sincerely thank the reviewer for their valuable insights and suggestions.

REFERENCES

1. Sharma, S. Nanocarrier mediated delivery of insecticides into tarsi enhances stink bug mortality. *Nature Communications* **15**, 9737 (2024).
2. Symonds, B. L., Thomson, N. R., Lindsay, C. I. & Khutoryanskiy, V. V. Rainfastness of Poly(vinyl alcohol) Deposits on *Vicia faba* Leaf Surfaces: From Laboratory-Scale Washing to Simulated Rain. *ACS Appl. Mater. Interfaces* (2016).
3. Tang, J. Deposition and water repelling of temperature-responsive nanopesticides on leaves. *Nature Communications* **14**, 6401(2023).
4. Wan, M. *et al.* Degradable ZIF-8/silica carriers with accropode-like structure for enhanced foliar affinity and responsive pesticide delivery. *Chemical Engineering Journal* **489**, 151301 (2024).
5. Chen, J. Y., Kuruparan, A., Zamani-Babgohari, M. & Gonzales-Vigil, E. Dynamic changes to the plant cuticle include the production of volatile cuticular wax-derived compounds. *Proc. Natl. Acad. Sci. U.S.A.* **120**, e2307012120 (2023).
6. Gudesblat, G. E., Torres, P. S. & Vojnov, A. A. *Xanthomonas campestris* Overcomes Arabidopsis Stomatal Innate Immunity through a DSF Cell-to-Cell Signal-Regulated Virulence Factor. *Plant Physiol.* **149**, 1017–1027 (2009).
7. Bianco, M. I. *et al.* Xanthan Pyruvilation Is Essential for the Virulence of *Xanthomonas campestris* pv. *campestris*. *MPMI* **29**, 688–699 (2016).